# Analysing stress field conditions of the Colima Volcanic Complex (Mexico) by integrating FEM simulations and geological data

Silvia Massaro[1,2], Roberto Sulpizio[1,2,3], Gianluca Norini[2], Gianluca Groppelli[2], Antonio Costa[1], Lucia Capra[4], Giacomo Lo Zupone[5], Michele Porfido,[6] Andrea Gabrieli[7]

[1]Istituto Nazionale di Geofisica e Vulcanologia, Via D. Creti 12, 40128, Bologna, Italy.
[2]Istituto di Geologia Ambientale e Geoingegneria, Consiglio Nazionale delle Ricerche, Via M. Bianco 9, 20131, Milan, Italy.
[3]Dipartimento di Scienze della Terra e Geoambientali, Via E. Orabona 4, 70125, Bari, Italy.
[4]Centro de Geociencias, Universidad Nacional Autonoma de Mexico, Queretaro, Mexico.
[5]Institute of New Energy and Low-carbon Technology, Sichuan University, Chengdu, PRC.
[6]Alumni Mathematica, Dipartimento di Matematica, Via E. Orabona 4, 70125, Bari, Italy.
[7] Hawai'i Institute of Geophysics and Planetology, 1680 E-W Road, Honolulu, Hawai'i 96922, USA.

*corresponding author: Silvia Massaro (silvia.massaro@ingv.it)

## Abstract

In the last decades numerical methods have become very popular tools in volcanological studies, since capable of considering many relevant parameters in their calculations, such as the presence of multiple reservoirs, topography, and heterogeneous distribution of host rock mechanical properties. Although the widespread availability of geodetic data is keep growing, the influence of geological data on the numerical simulations is still poorly considered. In this work a 2D Finite Element Modelling is provided by using the LInear Static Analysis (LISA) software, in order to investigate the stress field conditions occurring around the Colima Volcanic Complex (CVC, Mexico) at increasing the details of geological and geophysical input data. By integrating the published geophysical, volcanological, and petrological data, we provide a first-order description of the domain of the CVC feeding system, considering either one or two magma chambers connected to the surface via dykes or isolated (not connected) in the elastic host rocks. We test the methodology by using a gravitational modelling with different geometrical configurations and constraints (i.e. magma chamber dimensions, depth, overpressure). Our results suggest that an appropriate set of geological data is of pivotal importance for improving the mesh generation procedures and the degree of accuracy of numerical outputs, aimed to more reliable physics-based representations of the natural systems.

## 1 Introduction

Magmatism and tectonism in volcanic active areas are strongly related to the regional and local stress fields, affecting both the orientation of faults and the location of volcanic vents, two fundamental

aspects when interpreting volcanic unrest and forecasting volcanic eruptions (Geyer et al., 2016). The stress field around a magmatic source originates from three main contributions: (1) the background stress, composed of a vertical gravitational load and a lateral horizontal load corresponding to lithostatic confinement and tectonic regimes; (2) the stress field caused by the loading of the volcano edifice; and (3) the stress field generated by the magmatic overpressure in the chamber system (e.g. Martí and Geyer, 2009; Currenti and Williams et al., 2014). In recent years, a large number of semi-analytical and numerical solutions for the stress field state of geological and volcanological systems have been proposed (e.g. Cayol and Cornet, 1998; Simms and Garven, 2004; Manconi et al., 2007; Long and Grosfils, 2009; Currenti et al., 2010; Currenti and Williams et al., 2014; Zehner et al., 2015), taking into account the static elastic deformation in a multi-layered half-space (e.g. Dieterich and Decker, 1975; Bonafede et al., 2002; Wang et al., 2003; Gudmundsson and Brenner, 2004; Zhao et al., 2004; Pritchard and Simons, 2004; Gottsmann et al., 2006; Geyer and Gottsmann, 2010; Zhong et al., 2019). Following the successful application in mechanical engineering, the use of Finite Element Method (FEM) has been extensively introduced in Earth Sciences in order to investigate the effects of topography, lithologic heterogeneities, tectonic stresses and the gravity field on the Earth's surface deformation (e.g. Cailleau et al., 2003; 2005; Buchmann and Conolly 2007; Manconi et al., 2009; Masterlak et al., 2012), including volcanoes (e.g. Fujita et al., 2013; Carcho and Gàlan del Sastre, 2014; Bunney, 2014; Ronchin et al., 2015; Hickey et al., 2015; Cabaniss et al., 2019; Rivalta et al., 2019).

The use of FEM in volcanic areas has several examples, which vary from the influence of layered materials on the surface deformation process during volcanic inflation (e.g. Darwin volcano, Galapagos Islands; Manconi et al., 2007; Albino et al., 2010) to processes affecting chamber rupture (e.g. Grosfils, 2007; Long and Grosfils, 2009). FEM is also used in fluid dynamics and thermodynamics (e.g., Gutiérrez and Parada, 2010; Gelman et al., 2013) for solving issues related to motion of fluids and heat transfer.

The local stress around a volcanic feeding system strongly depends on the magma chamber geometry and on the mechanical properties of the layered host rock around it (e.g. Martì and Geyer, 2009), mainly due to broad changes in Young's modulus (e.g. Gudmundsson et al., 2011; Jeanne et al., 2017; Heap et al., 2020). For instance, limestones, lava flows, welded pyroclastic units and intrusive

rocks can be very stiff (high Young's modulus; from ca. 1.7 to 27 GPa for limestones, Touloukian, 1981; ca. 5.4 GPa for volcanic rocks, Heap et al., 2020), whereas young and non-welded pyroclastic units may be very soft (low Young's modulus; ca. 1.7 – 3.1 GPa, Margottini et al., 2013). Consequently, the local stress may change abruptly from one layer to another (e.g., Gudmundsson, 2006). Irrespective of the scope of the numerical investigation, the importance of applying accurate rheological constraints to FEM modelling was discussed in many studies (e.g., Folch et al., 2000; Newman et al., 2001; Fernandez et al., 2001; Currenti et al., 2010; Geshi et al., 2012). This implies that geology of the volcanic area needs to be considered as more accurate as possible. However, few investigations have been carried out to assess the influence of the amount and quality of geological data into FEM computations (Kinvig et al., 2009; Norini et al., 2010, 2019; Cianetti et al., 2012; Ronchin et al., 2013; Chaput et al., 2014). To bridge this gap, in this work we use the Linear Static Analysis (LISA) software (version 8.0; www.lisafea.com) to study the subsurface stress behaviour in an elastic domain at Colima Volcanic Complex (CVC, Mexico) when improving the description of geological constraints.

The CVC area is a good candidate for testing the response of FEM software to different geological conditions, being constituted by a large volcanic complex (significant topographic load; Lungarini et al., 2005), a well-defined feeding system inferred from geophysical and petrological data (e.g. Spica et al., 2017; Massaro et al. 2018, 2019), and growth within a tectonic graben (bordered by normal faults; Fig. 1a) infilled by volcaniclastic material (variability of rock mechanical characteristics; Norini et al., 2010, 2019).

In this light, the present study proposes a contribution to a more proper use of FEM models for assessing the stress state pattern in volcanic areas at different levels of description of the geological features. In particular, we focus on the CVC by using the available published data of the inferred feeding system structure, in order to assess how the addition of geological and volcanological constraints (i.e. stratigraphy, geometry of the plumbing system, extensional tectonic regime, local fault systems) may, and at what extent, affect the model outputs (Fig 1b). Beside and beyond the evaluation of geological details on FEM outputs, we also obtained a picture of the large-scale stress distribution in the CVC subsurface.

## 2 The Colima Volcanic Complex (Mexico)

*2.1 Geological framework*

The Pleistocene-Holocene CVC is one of the most prominent volcanic edifices within the Trans-Mexican Volcanic Belt (TMVB) (Macías et al., 2006; Capra et al., 2016; Norini et al., 2019; Fig. 1a). In this area, the Rivera microplate and the Cocos plate subduct beneath the North America plate along the Middle American Trench, producing great deformation and fragmentation of the continental plate (Stock and Lee, 1994), and forming a triple junction that delimits the tectonic units known as the Jalisco Block (JB) and the Michoacán Block (MB) (Luhr et al., 1985; Allan, 1986; Rosas-Elguera et al., 1996; Rosas-Elguera et al., 1997; Ferrari and Rosas- Elguera, 1999; Rosas-Elguera et al., 2003; Frey et al., 2007). The three rifts of this system are the Tepic-Zacoalco (TZR), the Chapala-Tula (CTR), and the Colima Rift (CR) where the CVC is emplaced (Allan, 1986; Escudero and Bandy, 2017). The still active NS trending Colima Rift (CR) was formed during an extensional phase occurred after the Late Cretaceous–Paleogene compressive and transpressive phase (Allan, 1986; Serpa et al., 1992; Bandy et al., 1995; Cortés et al., 2010). The rifting phase deformed Cretaceous marine limestones, Jurassic–Tertiary metamorphosed clastic and volcaniclastic sediments, Cretaceous–Tertiary intrusive rocks and Tertiary-Quaternary volcanic deposits along sub-vertical crustal faults. While opening, CR was gradually filled with Pliocene–Quaternary lacustrine sediments, alluvium and colluvium (e.g. Allan, 1986; Allan et al., 1991; Norini et al., 2010). The geometry, kinematics and dynamics of the CR have been studied on the basis of field, seismic, and geodetic data, mainly collected in its northern and central sectors (see Fig. 1 in Norini et al., 2010).

The amount of vertical displacement of the northern and central sectors is estimated to be at least 2.5 km by adding the topographic relief of the bounding fault scarps (1.5–1.6 km) to the calculated sediment depth (Allan, 1985; Serpa et al., 1992). Field data and focal mechanism solutions are consistent with a direction of opening of the northern and central sectors oriented from E-W to NW-SE, with a mainly normal and minor right-lateral displacements of the bounding faults (Barrier et al., 1990; Suárez et al., 1994; Rosas-Elguera et al., 1996; Garduño-Monroy et al., 1998; Norini et al., 2010, 2019). In contrast to field and seismic evidence of long-term slightly dextral oblique extension, recent GPS geodetic measurements suggest a possible sinistral oblique extension of the CR (Selvans et al., 2011). In both cases, the stress regime is mainly extensional, with an approximately E-W

orientation of the minimum horizontal stress in the basement of the CVC (Barrier et al., 1990; Suárez et al., 1994; Rosas-Elguera et al., 1996; Selvans et al., 2011; Norini et al., 2010, 2019).

The CVC stands within the central sector of the CR, on top of the Cretaceous limestones, Late Miocene-Pleistocene volcanic rocks, and Pliocene-Holocene lacustrine sediments, alluvium, and colluvium (Allan, 1985, 1986, 1991; Cortès, 2005; Norini et al., 2010). The volcanic complex is affected and displaced by the N-S/NNE-SSW-trending recent-active crustal faults of the CR, controlling the geometry and location of the volcano feeding system (Fig. 1a). Indeed, the CVC was formed by three andesitic stratovolcanoes aligned parallel to the CR bounding faults: the northern inactive Cantaro volcano (2900 m a.s.l.), following by the inactive Nevado de Colima (4255 m a.s.l.) and, in the southern part, the youngest and active Volcán de Colima (3763 m a.s.l.) (Norini et al., 2019 and reference therein).

*2.2 Eruptive activity*

The eruptive history of the CVC started in the northeast area with the formation of Cantaro volcano at ca. 1-1.5 Ma. The volcanic activity of the Nevado de Colima started at ca. 0.53 Ma. It is composed of voluminous andesitic lava domes and flows and pyroclastic deposits associated with caldera forming eruptions and numerous partial sector collapses (Robin et al., 1987; Roverato et al., 2011; Roverato and Capra, 2013; Cortès et al., 2019). The youngest Volcán de Colima, now considered one of the most active volcanoes of the world, consists of the Paleofuego edifice that suffered several sector collapses, with the formation of a horseshoe-shaped depression where the new active cone (also known Volcán de Fuego) grew up, through Merapi and Soufrière type dome collapses, extrusion of lava flows, Vulcanian and occasionally sub-Plinian explosive eruptions (Saucedo et al., 2010; Massaro et al., 2018, 2019). The activity of both Nevado and Volcán de Colima volcanoes also included several sector collapses, occurred frequently in the Upper Pleistocene and Holocene, repeatedly devastating the floor of the Colima Rift down to the Pacific Ocean (Robin et al., 1987; Luhr and Prestegaard, 1988; Stoopes and Sheridan, 1992; Capra and Macias, 2002; Cortès, 2005; Roverato et al., 2011).

## 2.3 The CVC plumbing system

Spica et al. (2017) indicate a 15 km-deep low velocity body (LVB) as the CVC deep magma reservoir. Its horizontal extension seems to be delimited by the borders of the CR, suggesting a structural control of the normal fault system on it (Spica et al., 2014). The LVB has an extent of ca. 55 km × 30 km in the N-S and E-W directions respectively, showing a mean thickness < 8 km. Escudero and Bandy (2017) obtained a higher resolution tomographic image of the subsurface in the CVC area, showing that the most active magma generation zone is presently under the Fuego de Colima edifice. Here, the ambient seismic noise tomographic study proposed by Spica et al. (2014) confirmed the presence of a shallow magma chamber above ca. 7 km depth, as also demonstrated by petrological studies (Medina-Martinez et al., 1996; Luhr, 2002; Zobin et al., 2002; López-Loera et al., 2011; Reubi et al., 2013, 2019; Macìas et al., 2017). Cabrera-Gutiérrez and Espíndola (2010) suggested the shallow active magma storage has a volume of ca. 30 km$^3$. The shallow magma chamber is connected to the surface by a dyke/conduit system, whose path is facilitated by the presence of the CR fault zone, which provides a natural pathway for fluids (e.g., Allan, 1986; Norini et al., 2010, 2019). The arrangement of dykes and the alignment of volcanic centres of CVC suggest that the dykes swarm draining the magma chambers developed along the NNE-SSW-trending, steep, eastward dipping normal fault exposed on the northern CVC flank (Norini et al., 2010, 2019).

Taking into account the previous information, Massaro et al. (2018) provided a first-order geometrical reconstruction of the Fuego de Colima feeding system during the 1913 sub-Plinian eruption, by using volcanological data (Saucedo et al., 2010, 2011; Bonasia et al., 2011) as input and constraints for numerical simulations. Results showed good matches for a hybrid configuration of the shallow conduit-feeding system (i.e., dyke developing into a shallower cylindrical conduit). The best-fit dyke geometry has width in the range from 200 m to 2000 m and thickness of ca. 40 m, with the cylindrical conduit diameter similar to the dyke thickness. The shallow magma chamber top was set at 6 km of depth, and dyke-cylinder transition at 500 m below the summit, as also inferred from geophysical data (Salzer et al., 2014; Aràmbula et al., 2018).

## 3 Methods

In this study, we used the commercial 8.0 version of LISA software (www.lisafea.com). LISA is a general-purpose Finite Element Analysis (FEA) software developed in the '90s based on the formulations proposed by Rao (1989). Since then, formulations from many other sources were also integrated (Bathe, 1990; Michaeli, 1991; Schwarz, 1991; Babuska et al., 1995). Despite FEA was originally used for structural analysis (Rao, 1989; 2013), it is also able to successfully predict the stress-strain behaviour of rock masses accounting for elastic models, in particular the deformation and failure mechanisms even in layered rock masses (Gabrieli et al., 2015).

Simplifying techniques in structural FEA can give valuable insights into local stresses more rapidly and efficiently than a full 3D model. Here we considered a 2D model throughout a complex structure (i.e. dual magma chamber feeding system, rift system, rock layering, and faults), in order to investigate the stress behaviour induced in the host rocks in response to the increasing detail of geological data used to constrain the model.

*3.1 Modelling approach*

Taking into account the works of Norini et al. (2010, 2019), we simulated the stress field of the CVC plumbing system considering an E-W cross-section, which is parallel to the extension associated to the active Colima Rift (Norini et al., 2010), shown in Figure 1a-b (a-a').
Since the extent of the CVC magma chambers in the NNE-SSW direction is typically much longer than the dimensions of the E-W cross section (Spica et al., 2017), 2D solutions of either numerical or analytical models describing E-W elongated magma chambers in the crust can be reasonably adopted (Jaeger et al., 2009; Costa et al., 2011). A topographic profile and 2D plane along the chosen E-W cross-section of the CVC area was obtained in ESRI ArcGIS from a Digital Elevation Model (DEM; resolution 50 m; Instituto Nacional de Estadística y Geografía - INEGI https://en.www.inegi.org.mx/). This cross section was imported into Autodesk Auto-Cad R13 and approximated to a third-degree spline. Finally, the IGES file was imported into LISA, where the mesh discretization was performed. The domain was discretized by three and four-node finite elements (Table 1; Fig. 1c). The volcanic

area domain extends 60 km horizontally and 30 km below the surface set in an *x-z* Cartesian
Coordinate System. Zero normal displacements are assigned at the bottom and the lateral boundaries
of the domain, while the upper boundary representing the ground surface is stress free (Fig. 1c). The
analysis is carried out by using a plane strain approximation, implying that the deformation in the
third direction is assumed to be negligible.
FEM of geological structures requires accurate discretization of the computational domain such that
geological units are represented correctly. Zehner et al. (2015) reported that the unstructured
tetrahedral meshes on a complex geological model has to fulfil the following requirements: i)
sufficient mesh quality: the tetrahedrons should not be too acute-angled, since numerical instabilities
can occur, ii) incorporation of geometry for defining boundary conditions and constraints, iii) local
adaption, which is a refinement of the mesh in the vicinity of physical sources in order to avoid
numerical errors during the simulation. Considering these requirements, in this work we adopt as the
best discretization a mesh with 4660 plane continuum elements for the E-W cross-section. The size of
finite elements was refined in the regions with higher gradients, especially near the contours of the
magmatic feeding systems.
In our simulations, the extent of the rock layers (Table 2) is referred to the model of Norini et al.
(2010, 2019). Magma chambers and dykes are considered as pressurized finite-size bodies in an
elastic crustal segment, acting as fluid-filled holes. The boundary condition (pressurization) is
provided by applying internal forces that act on the walls. This approach has been extensively used in
several analytical and numerical models that treat magma reservoirs as internally pressurized
ellipsoidal cavities within an elastic half space, in order to gain insight into the behaviour of magma
plumbing systems (Pinel and Jaupart, 2004; Gudmundsson, 2006; Grosfils, 2007; Andrew and
Gudmundsson, 2008; Hautmann et al., 2013; Currenti and Williams, 2014; Zhong et al., 2019).
The geometrical configuration set for the CVC feeding system (i.e. the shape and dimensions of the
magmatic chambers) derives from the literature (Spica et al., 2014, 2017; Massaro et al., 2018, 2019)
and it is simplified in Figure 1d. The overpressure in magma chambers may be produced by a variety
of processes, including fractional crystallization, volatile exsolution and magma recharge, leading to
deviatoric stresses in the country rock that may be tens of MPa in magnitude (Jellinek and DePaolo,
2003; Karlstrom et al., 2010).
Previously published studies indicate that differences between, and problems with, elastic models
derive principally from the key role played by gravity (e.g. Albino et al., 2018). Gravity plays a first
order role on bedrock failure conditions (Gerbault, 2012), on the geometry of magma propagation
with respect to an edifice load and on buoyancy contrasts driving magma upward (Lister and Kerr,
1991; Watanabe et al., 2002). However, in a wide variety of simulations of natural phenomena the
gravitational effects are often incorporated either incorrectly or incompletely (e.g. Grosfils, 2007).
Some authors argued on whether it is appropriate or not to account for the gravity body force in
numerical models of volcanic inflation (e.g. Currenti and Williams, 2014; Grosfils et al., 2015).
When the gravitational loading is not included in the model, the volcanic deformation results from a
change with respect to a stage previously at equilibrium (e.g. Gerbault et al. 2018). In this work, we
carried out simulations considering the effect of the gravitational loading. Gravity in the host rock is
implemented via body forces. The model initial condition has a pre-assigned lithostatic stress, whose
computation, in presence of topography and material heterogeneities, is not trivial because it requires
applying the gravity load preserving the original not deformed geometry of the mesh (Cianetti et al.,
2012). Since the presence of a lithostatic stress field, the load applied at the reservoir boundaries
represents a superposition of the magmatic overpressure and lithostatic component.
We also took into account the effect of the existing faults of the Colima Graben (CG) system even if
LISA cannot include a frictional law to represent the fault movement (i.e. Chaput et al., 2014). As
reported in Jeanne et al. (2017 and reference therein) the damage induced by faults increases from the
host rocks to the fault core implying the reduction in the effective elastic moduli represented by a
progressive decrease in Young's Modulus. Considering the evaluation of fault zone elastic properties
provided by Jeanne et al. (2017), we represented the faults bordering the CG as two damage zones
inclined of ca. 70° and with a thickness of ca. 1 km, showing reduced elastic properties with respect
to the surrounding host rocks down to 10 km in depth.
It is important to note that we chose to represent the different simulations using different colour
scales. Although such a choice makes more difficult a visual comparison of the simulation outputs
and it needs to be kept in mind looking at the different figures, it preserves the necessary details of
stress distribution, which would have been lost using a common colour scale for all the figures in
LISA.

**4 Geological data**
In this work, we used geological information available in literature as input data, in order to estimate
the stress variations around the CVC magmatic plumbing system. Here we briefly describe the main
geological features taken into account in LISA simulations.
*4.1 Stratigraphy*
Four units forming the CVC system were defined from the available geological data (Table 2): i)
Basement (Unit B): cretaceous limestones and intrusive rocks forming the bed-rock underlying the
CVC; ii) Graben fill deposits (Unit GF): Quaternary alluvial, colluvial, and lacustrine deposits filling
the graben; iii) Fuego de Colima deposits (Unit FC): andesitic lavas and pyroclastic deposits forming
the Paleofuego-Fuego de Colima edifices; and iv) Volcaniclastic deposits (Unit VD): volcaniclastic
deposits covering the southern flank of the CVC (e.g. Cortés et al. 2010; Norini et al., 2010, 2019).
Being the area interested by FEM extended down to 30 km, it is evident how Unit B is dominant with
respect to the others, which occupy only few km in the upper part of the simulated domain. We
assumed constant mechanical characteristics within each Unit (Table 2). In particular, Unit B was
considered mechanically homogeneous with elastic properties of a carbonate, due to the lack more
detailed information of deeper lithologies (Norini et al., 2019).
Deformation within the brittle upper crust is described by elastic material behaviour. For each Unit
we fixed typical rock mass properties, density ($\rho$), Young's Modulus (E) and Poisson's Ratio ($\nu$)
(Table 2). The rock masses are considered dry, in order (eventual) pore pressure to be neglected.
Only for Unit GF a higher value for the Poisson's Ratio was used close to the surface in order to
mimic high water content in the graben sediments. The maximum thickness of the graben fill (about
1 km) was assumed from the literature (Allan, 1985; Serpa et al., 1992; Norini et al., 2010, 2019). For
Units B and GF rock mass proprieties were derived from Hoek and Brown (1997) and Marinos and
Hoek (2000), while for volcanic materials (units FC and VD; Table 2) were estimated according to
the approach proposed by Del Potro and Hürlimann (2008). This information allowed Norini et al.
(2019) to derive the equivalent Mohr-Coulomb properties for the stress ranges expected in the
different sectors of the CVC. In addition, in order to describe the effects of the CG faults on stress
field distribution, the mechanical properties were locally degraded in proximity of the faults
themselves.
*4.2 The geometry of the plumbing system*
The geometry of the E-W cross-section of the CVC plumbing system was modelled taking into
account the previous subsurface information described in Section 4.1. In our 2D model, we assumed
the CVC composed of a two magma chambers connected by dykes and to the surface by a conduit
(Fig. 1d). The shape of the magma chambers and dykes are represented by elliptical cross-sections
with the major (*2a*) and minor (*2b*) axes.
Generally, the magma chambers have a sill-like shape that is often imaged in seismic studies of
volcanoes and rift zones (Macdonald, 1982; Sinton and Detrick, 1992; Mutter et al., 1995; MacLeod
and Yaouancq, 2000; Singh et al., 2006; Canales et al., 2009). Most of them are not totally molten but
rather a mixture of melt and crystal mush (i.e. Parfitt and Wilson, 2008). Various estimates have been
made to infer the actual amount of melt in a magmatic body, showing that it is only ca. 10% of the
total chamber volume (Gudmundsson et al., 2012 and reference therein).
Spica et al. (2017) described a 15 km-deep low velocity body (LVB) with its top at ca. 15 km of
depth and with an estimated volume of ca. 7000 $km^3$, representing the deep magmatic reservoir of
CVC. Assuming the melt as 10%, the deep magma chamber volume would be ca. 700 $km^3$.
Simplifying this volume in an elliptical sill-like geometry, the dimensions (i.e. *2a, 2b, 2c* axes) have
to be scaled according to those of LVB (55 × 30 × 8 km; Spica et al., 2017). We therefore fixed *2a* =
14 km, *2b* = 3.6 km, *2c* = 26 km as the dimensions of the deep magma chamber, being *2c* elongated
in NW-SE direction.
For the shallow part of the feeding system, we have no detailed geophysical constraints. However,
Massaro et al. (2019) reproduced through numerical modelling the nonlinear cyclic eruptive activity
at Fuego de Colima in the last 20 years, using a shallow magma chamber volume in the range of 20-
50 $km^3$, also according to the estimation of Cabrera-Gutiérrez and Espindola (2010). Assuming a
volume of 30 km$^3$, we fixed *2a* = 3.5 km, *2b* =2 km, *2c* = 8 km as dimensions of the shallow magma
chamber.
Numerous theoretical and field studies have established that host rock stresses dictate the magma
pathways (e.g. Maccaferri et al., 2011; Gudmundsson, 2011). During ascent to the surface, the dykes
align themselves with the most energy-efficient orientation, which is roughly perpendicular to the
least compressive principal stress axis $\sigma_3$ (e.g. Gonnermann and Taisne, 2015; Rivalta et al., 2019),
providing the magma driving pressure remains small compared to the deviatoric stress (Pinel et al.,
2017; Maccaferri et al., 2019). This behaviour, however, can be modulated in the presence of
significant variations in fracture toughness of the surrounding rock due to stratification (Maccaferri et
al., 2010) or to old and inactive fracture systems (Norini et al., 2019). Although for oblate magma
chambers the propagation of dykes is most probable from the tip areas, in our simulations the
orientation of dykes is assumed vertical, because of the preferential pathways represented by the CR
fault planes (Spica et al., 2017).
Although, for decades, magma conduits were modelled as cylinders, because of easiness of their
mathematical treatment, geophysical data and field observations highlight the importance and
peculiarities of dykes in magma transport and hence the need to adopt more realistic geometries
(Costa et al., 2009; Hautmann et al., 2013; Tibaldi, 2015). It is important to stress that although all
cavities/inclusions in a medium modify the local stress field and concentrate stresses, the induced
perturbation depends mainly on the geometry of the cavity/inclusion (Savin, 1961; Boresi et al.,
1985; Tan, 1994; Saada, 2009). We set the dimensions of feeder dykes in agreement with Massaro et
al. (2018): deep dyke *2ad* = 2 km; shallow dyke *2a* varies from 1 km at bottom to 500 m in the upper
part of the volcano; width of both deep and shallow dyke *2bd* = *2b* = 100 m (Fig. 1d), although the
exact value of the latter is not crucial for the purposes of this study. Moreover, it is worth noting that
it is not the aim of this work to provide the conditions for the magma chamber rupture, being LISA
accounting only for the elastic regime. For these reasons, the selected magma overpressures ($\Delta P$)
acting on the magma reservoirs and dykes have to be less than the tensile strength of the rocks. We
therefore fixed $\Delta P$ at 10 MPa and 20 MPa for the 15 km-deep chamber, and 5 MPa for the 6 km-deep
one. For the dykes and conduit, the magmatic overpressure is fixed at 10 MPa in the deeper dyke and
5 MPa in the shallower dyke, except for the upper 500 m of the shallower conduit where overpressure
is set at 0.4 MPa.
To take into account the effect of both far field extensive regime and CG around the magma feeding
system, we applied a uniform extension at the lateral boundaries of the domain (as reported in Martì
and Geyer, 2009) of 5 MPa and included two damage zones with reduced rock elastic moduli and
density (i.e. E = 1 GPa, $\nu$ = 0.20; Jeanne et al., 2017; $\varrho$ = 1850 kg/m$^3$).


**5 Results**
The first part of this section is focused on a sensitivity analysis of Young modulus variation, aimed to
quantify the numerical effects of approximation of this important rock property on FEM outputs. The
second part of this section describes the model outputs when adding complexity to the input
geological/geophysical data.
Considering the E-W cross-section (a-a'; Fig. 1a), we provided six domain configurations with
increasing geological complexity: i) "*homogeneous lithology model*" in which the volcanic domain is
only composed of andesite rocks; ii) "*not homogeneous lithology model*" where different geological
units are considered; iii) "*single magma chamber model*" composed of a not homogeneous lithology
and a 15 km-deep magma chamber; iv) "*dual magma chamber model*" composed of a not
homogeneous and 6 km- and 15 km-deep magma chambers; v) "*conduit feeding system model*"
composed of not homogeneous lithology, 6 km- and 15 km-deep magma chambers connected
through a deep-dyke, and a shallow conduit connecting to the surface; vi) "*extensional model*", in
which we added a 5 MPa horizontal extensional stress (far field) and, vii) "*faulted model*", in which
two damaged zones mimicking the CG faults were added to the "*extensional model*" (local stress)
(Fig. 1b).
The number of nodes in the *only substratum* and *single magma chamber* models is set at 4426, for the
dual magma chamber model is set at 4161, and at 3737 for the *conduit feeding system* and *faulted*
models.
*5.1 Sensitivity analysis on selected input parameters*
In order to quantify the influence of Young Modulus selection on the model outputs, we performed a
sensitivity test using the single magma chamber model as reference case. We evaluated the influence
of varying the Young Modulus in each geological Units on the principal stresses $\sigma_1$ and $\sigma_3$. Taking
into account the material properties used in the simulations (Norini et al., 2010, 2019; Table 2) as
reference values, we compared the stress state of the computational domain at changing (±) Young
Modulus by an order of magnitude. This variation has been separately applied to each Unit, in order
to assess what is the effect of changing material properties on model outputs. This sensitivity analysis,
although incomplete, may lead to raise awareness on the selection of input data when running a FEM.
The sensitivity analysis was carried out on a reduced simulation domain (the *x*-axis was set to 35 km)
in order to diminish the influence of binding effects that are present along domain borders.
We used the Euclidean norm (L2) method for illustrating the results of the sensitivity analysis. The
L2 norm applied on a vector space *x* (having components *i = 1,...n*) is strongly related with the
Euclidean distance from its origin, and is equal to:

$||x||_2 = \sqrt{\sum_i^n xi^2}$                        (1)

In our case, the vector space *x* is composed of all nodes of the computational domain (Table 1). We
defined $x_{ref}$ the vector containing the results for the maximum and minimum principal stress when
using the selected values of material properties (Table 1) and *x(-)*, *x(+)* the vectors at varying the
Young Modulus of one order of magnitude in each Unit.
We evaluated the global variation of stress in the proposed geometrical configurations of the domain
(i.e. not homogeneous lithology, single magma chamber, dual magma chamber, and dual magma
chamber with conduits models) calculating the global relative variation in L2 as follow:

$L2(-) = \dfrac{||x_{ref} - x(-)||_2}{||x_{ref}||_2}$                (2)
$L2(+) = \dfrac{||x_{ref} - x(+)||_2}{||x_{ref}||_2}$                (3)
In Figure 2 are reported the global relative variations in L2 of the principal maximum stress $\sigma_1$ and
principal minimum stress $\sigma_3$ caused by the variation of Young's Modulus in each Unit. All the
geometric configurations show variability less than 15%, with few exceptions within Unit B that have
variability over 30% (Fig. 2). It is worth noting that the spatial distribution of the major variations
seems to not significantly affect the final stress distributions, because: i) they are located near the
mesh borders (Fig. 3a, b); and, ii) when not at the mesh borders, the variations are limited to few %
(Fig. 3c, d). It means that changing the Young's Modulus of one order of magnitude produces
variation in FEM outputs distributed over a large domain and the change affecting the single nodes is
limited to few %.

*5.2 Homogeneous and not homogeneous lithology*
We carried out LISA simulations considering the effect of the gravitational loading on the
homogeneous and not homogeneous lithology on FEM outputs. In Figure 4 we reported a gravity
loading model for E-W cross-section of the CVC system. We first considered the homogeneous rock
composition composed by only andesitic lavas (Fig. 4a) and then by carbonates (Unit B), alluvional,
volcaniclastic and pyroclastic deposits (Units GF and VD*;* Fig. 4b). We analysed the principal
stresses $\sigma_1$ and $\sigma_3$ acting on the system, which correspond to the maximum and minimum stress at
a point, respectively.
Figure 4 shows the patterns of the minimum principal stress $\sigma_3$ (panels i-ii) and of the maximum
principal stress $\sigma_1$ (panels iii-vi), highlighting very slight differences between the homogeneous and
not homogeneous lithology cases. It is very important to stress that the *x-z* zero displacement
assigned at the bottom and the lateral boundaries of the domain created substantial artefacts in the
results (i.e. curved patterns of stress). The artefacts are also evident when considering $\sigma_3$ (panels i-ii)
where the boundary effect on *x*-axis is amplified by the presence of the upper free surface. For this
reason, the only area to be considered as unperturbed is the central part of the entire domain, and it
extends ca. 30 km horizontally and ca. 15 km vertically (within the blue contour in Fig. 4).

*5.3 Gravitational modelling using the inferred feeding system geometry*
We progressively add the elements of the conduit/feeding system of the CVC to FEM under the
effect of the gravitational loading. Three cross-section profiles (Figs. 5, 6) show increasing
complexity of the feeding system starting from a single magma chamber, passing to two magma
chambers, then adding the conduits, and, finally, considering the effects of faults.
Figure 5a describes the distribution of the minimum principal stress $\sigma_3$ (panel i) and the maximum
principal stress $\sigma_1$ (panel ii) at magma chamber overpressure of 10 MPa, showing how the insertion
of the pressurized magma chamber modifies the lithostatic stress. No significant differences in
magnitude and pattern of stresses are visible when having a magma chamber overpressure of 20 MPa
(Appendix 1a).
The addition of the shallow magma chamber significantly changes the values and pattern of both $\sigma_3$
and $\sigma_1$ (Fig. 5b). In particular, $\sigma_3$ and $\sigma_1$ stresses describe a typical inflation pattern produced by
overpressurised magma chamber(s) (Anderson, 1936; Gudmundsson, 2006), producing well-defined
stress arches of $\sigma_3$ (red dotted lines in Figs. 5bi) and divergent strong gradients of $\sigma_1$, well developed
around the larger magma chamber (Fig. 5bii). Stress arch is a common phenomenon occurring in
continuous materials as response to applied pressure. It has been proved to have great influences on
the self-stabilization of soils or rock masses (Huang and Zhang, 2012), and may influence
mechanisms of caldera collapse (Holohan et al., 2015). Very slight differences in magnitude and
pattern of stresses appear when using 10 MPa (Fig. 5b) or 20 MPa of deep magma chamber
overpressure (Appendix 1b).
Figure 6 shows the effect of adding two conduits connecting the deep and shallow magma chambers.
It is evident how the insertion of the conduits in the feeding system of CVC dramatically changes the
stress distribution, with disappearance of the stress arch and an almost constant stress in the
computational domain except than on the tips of the deep magma chamber.

*5.4 Extensional field stress*
In order to explore the influence of the extensional far field stress on stress patterns (Fig. 1a), we run
simulations applying 5 MPa of extensional stress to the FEM domain, which is a typical low value for
rift zones (Turcotte and Schubert, 2002; Moeck et al., 2009; Maccaferri et al., 2014; Sulpizio and
Massaro, 2017; Fig. 7).
In the case of a single magma chamber (with 10 MPa overpressure; Fig. 7, panels i-ii), the addition of
extensional far field stress reduces the confinement effect due to the no displacement condition
imposed along the *x-z* directions (plane strain approximation). The effect of the extensional field
stress on double magma chamber configuration (with 10 MPa overpressure in the deep chamber and
5 MPa in the shallower one) produces slight changes in stress magnitude and pattern for both $\sigma_3$ and
$\sigma_1$ (Fig. 7, panels iii-iv) with respect to Figure 5b. The same applies also for the complete feeding
system configuration, in which the attrition of the far field stress slightly changes the intensity of the
stresses and patterns (Fig. 7, panels v-vi). Using 20 MPa overpressure in the deep magma chamber
does not significantly affect the model outputs (Appendix 2).

*5.5 Faults bordering the Colima Rift*
In order to reproduce the effect of faults bordering the Colima Rift on the final feeding system
configuration, we added two damage zones by degrading the elastic properties of a volume of rock
mass. The insertion of the two zones of weakness does not alter significantly the stress distribution
observed in Figures 7v and 7vi, with only reduction of both $\sigma_1$ and $\sigma_3$ values in the surroundings of
the damage zones (Figs. 7vii and 7viii). The different distance of the two damage zones to the
feeding system (especially the deep magma chamber) produces a small asymmetry in both $\sigma_1$ and $\sigma_3$
patterns with respect to simulations without damage zones (Figs. 7v-viii).


**6 Discussions**

*6.1 FEM analysis at increasing geological details*
The presented FEM model of the CVC highlighted some important characteristics of crustal stress
distribution at changing geological constraints used as input conditions (Spica et al., 2014, 2017;
Massaro et al., 2018). Although the results have to be considered as a first order approximation, the
changes in stress distribution are evident and useful for the understanding of limitations and
advantages of FEM.
Under the assumptions of plane strain, gravitational loading, and overpressured magma chambers and
dykes, the use of homogeneous or not homogeneous lithology for FEM provides negligible effects in
stress intensity and pattern (Fig. 4). This is because the upper Units (Units FC, VD, GF; Table 2)
represent only a limited part of the simulated domain, which in the remaining part results entirely
composed of the assumed homogeneous basement (Unit B; Table 2). This does not mean that the
influence of the upper Units may be still negligible using smaller scales of the simulated domain.
Analysing the FEM outputs with the single magma chamber, it emerges how the overpressures, $\Delta P$,
only limited the effects of gravitational loading. The use of a dual magma chamber geometry better
describes the inflation induced by overpressure within magma chambers, with the formation of the
stress arch in the minimum compressive stress $\sigma_3$ plot. It is important to highlight that for both single
and dual magma chamber models, the change of internal overpressure from 10 to 20 MPa slightly
changes the magnitude of the stress but not their general patterns (Appendix 1-2).
The presence of dykes in the magma feeding system dramatically change the $\sigma_3$ and $\sigma_1$ patterns (Fig.
6). Indeed, they become quite homogeneous throughout the computational domain, with the only
exception of sidewall effects induced by the zero displacement conditions, already discussed in
Figure 4.
The addition of extensional field stress of 5 MPa reduces the sidewall effects and produces an almost
homogeneous stress distribution in the upper part of the computational domain, above the top of the
deep magma chamber. This, along with the additional inclusion of the damage zones introduced to
mimic the effects of CG faults, describes a close to equilibrium volcanic system, in which volcanic
overpressure and lithostatic stress almost equilibrate each other (Sulpizio et al., 2016).

*6.2 Some implication of the stress state of the CVC inferred from FEM*
The results obtained with the insertion of the full feeding system and far field stress on the FEM
highlight an almost homogeneous stress distribution in the CVC area. This means that the shape of
the dual magma chamber feeding system model and far field stress provide a stable geometry, which
limits the stress changes to few MPa. All the large stress variations are located at the tips of the

magma chambers, as expected for pressurized or under-pressurized cavities in the lithosphere (Martì and Geyer, 2009). This means that the whole feeding system is in a quasi-equilibrium state, and, as an example, any overpressure created by input of new magma is adjusted by increasing the magma chamber volume or erupting at the surface. Even if we consider the scenario of complete emptying the upper conduit and part of the shallow magma chamber, as occasionally occurred during the past sub-Plinian and Plinian eruptions (Luhr et al., 2002; Saucedo et al., 2010; Massaro et al., 2018), this would result in the restoration of the stress arch, which is still a stable stress configuration. Even the complete emptying of the shallow magma chamber probably would be ineffective for triggering a large collapse (caldera forming) of the feeding system. This latter event would be possible only if a large depressurization of the deeper magma chamber would occur, but it implies the eruption of tens to hundreds of $km^3$ of magma, which seems not very likely provided the current stress distribution in CVC.

Beside and beyond the limitations due to the first order approximation of the FEM analysis, other sources of uncertainties in the discussion about present and future stress state of the CVC come from not considering gravity-driven processes, such as volcano spreading due to plastic deformation of the GF Unit (Norini et al., 2010, 2019) or pressurization of the shallower conduit (Massaro et al., 2018), and detailed regional tectonics (Norini et al., 2010, 2019). The effect of the two fault systems bordering the Colima Rift were simulated by degrading the mechanic properties of rocks in an area of about 1 km width up to a depth of 10 km. Although the effects are negligible at the scale of the computational domain, it cannot be excluded some local significant effects that cannot be resolved using the described approach.

## 7 Conclusions

The increasing details of geological and geophysical data to FEM simulation at Colima Volcanic Complex (Mexico) showed the importance of using the most accurate input data in order to have reliable outputs. In particular, the data here presented highlighted how the use of simplified models produces unreliable outputs of the stress state of the volcano subsurface.

Beside and beyond the results obtained by analysing the influence of detailing geological and
geophysical data, the FEM of CVC confirms the close to equilibrium state of the volcano, which is
the expected stress distribution induced by a feeding system directly connected to the surface.
This means that any overpressure created by input of new magma is adjusted within the feeding
system, sometimes triggering eruptions. The complete emptying the upper conduit and part of the
shallow magma chamber, as occasionally occurred in the past, originating sub-Plinian and Plinian
eruptions, would result in the restoration of the stress arch, which is still a stable stress configuration.
Descends that large magnitude, caldera forming eruptions are possible only if the bigger deep magma
chamber is also involved and significantly emptied during an eruption.

**Appendices**

**Appendix 1**
E-W gravitational modelling of the CVC domain (stratified lithology) for all configurations
investigated. The magnitude and pattern of the principal stress account for a) single magma chamber
model (number of nodes: 4426); b) dual magma chamber model (number of nodes: 4161); c) dual
magma chamber with conduits model (number of nodes: 3737). The dimension of the deep magma
chamber: $2a$ = 14 km and $2b$ = 3.6 km at 15 km of depth; shallow magma chamber: $2a$ = 3.5 km and
$2b$ = 2 km at 6 km. The magmatic overpressure is 20 MPa for the deep chamber, and 5 MPa for the
shallower. Black dotted lines highlight the passage from different stress values. Note that the scale of
stress values are different for each panel in order to maximise the simulation details.

**Appendix 2**
E-W gravitational modelling of the CVC domain (stratified lithology) considering a far extensional
stress field of 5 MPa for all configurations investigated. The magnitude and pattern of the principal
stress account for a) single magma chamber model model (number of nodes: 4426); b) dual magma
chamber model (number of nodes: 4161); c) dual magma chamber with conduits model (number of
elements: 3737).  The dimension of the deep magma chamber: $2a$ = 14 km and $2b$ = 3.6 km at 15 km
of depth; shallow magma chamber: $2a$ = 3.5 km and $2b$ = 2 km at 6 km. The magmatic overpressure
is 20 MPa for the deep chamber, and 5 MPa for the shallower. Black dotted lines highlight the
passage from different stress values. The red arrows indicate the direction of the applied far field
stress. Note that the scale of stress values are different for each panel in order to maximise the
simulation details.

**Code/Data Avaiability**
The LISA code is available at https://lisafea.com/.

**Author's contribution**
SM, RS, AC, GN and GG conceived the study. SM and RS wrote the bulk of the manuscript with the
input of all the co-authors. SM and GL compiled the numerical simulations and formulated the
adopted methodology. MP and SM carried out the sensitivity analysis. RS, AC, SM, GN, GG, LC,
GL, MP and AG worked on the interpretation of the results.

**Competing interests:** The authors declare that they have no conflict of interest.
**Acknowledgements:** SM thanks the LISA customer service for the support received.

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

L047

L048 **Table 1 -** Element types used in LISA analysis considering the final conduit feeding system
L049 configuration – Fig.1d, panel vi)

| *E-W cross-section (a-a')* | | Element Type | Elements | Nodes |
|---|---|---|---|---|
| L050 | | | | |
| L051 FC | Fuego de Colima | quad4-tri3 | 372 | 384 |
| L052 VD | Volcanic Deposits | quad4-tri3 | 245 | 273 |
| L053 GF | Graben Fill | quad4-tri3 | 456 | 338 |
| L054 B | Basament | quad4-tri3 | 3088 | 2907 |
| L055 CG | Colima graben | quad4-tri3 | 48 | 71 |

L056 Total Elements: 4209

L057 **Table 2 -** Rock mass and mechanical properties of the geological Units used in the finite-element
L058 model (from Norini et al., 2010, 2019).
L059

| Acronym | Model Unit | Rock Type | Density (kg/m$^3$) | Young's Modulus (MPa) | Poisson's ratio $v$ |
|---|---|---|---|---|---|
| FC | Fuego de Colima | Andesitic lavas and pyroclastic deposits forming the Paleofuego-Fuego de Colima | 2242 | $1.4 \times 10^3$ | 0.30 |

| | | | | | |
|---|---|---|---|---|---|
| | | volcano | | | |
| VD | Volcaniclastic deposits | Pyroclastic and epiclastic deposits covering the southern flank of the CVC | 1539 | $1.7 \times 10^3$ | 0.32 |
| GF | Graben Fill | Quaternary alluvial, colluvial, lacustrine deposits filling the graben | 1834 | $1.5 \times 10^3$ | 0.35 |
| B | Basement | Cretaceous limestones and intrusive rocks forming the bed-rock underlying the CVC | 2650 | $3.6 \times 10^4$ | 0.30 |

L060

L061

L062 **Figures Captions**

L063

L064 **Fig. 1** (a) Morphotectonic map of the Colima Volcanic Complex (NC=Nevado de Colima volcano;
L065 FC=Fuego de Colima volcano) and Colima Rift with the main tectonic and volcano-tectonic
L066 structures (NCG =Northen Colima Graben; CCG= Central Colima Graben, from Norini et al., 2019).
L067 In the inset, the location of the Colima Volcanic Complex (CVC) within the Trans-Mexican Volcanic
L068 Belt (TMVB) is shown in the frame of the subduction-type geodynamic setting of Central America
L069 (from Davìla et al., 2019); (b) general sketch of the geometrical configurations used in LISA; (c)
L070 example of mesh of the investigated area for the dual magma chamber model with conduits (case v in
L071 panel (b), considering zero-displacement along the bottom and left and right sides. Note that for case
L072 (vi) in panel (b) the zero-displacement is removed from the lateral sides; (d) sketch of the Fuego de
L073 Colima feeding system composed of a 15 km-deep magma chamber connected to surface via a 6 km-
L074 deep magma chamber and dykes. $\Delta P_{chs}$ and $\Delta P_{chd}$ are the magmatic overpressures in the shallow
L075 and deep chambers, respectively (modified from Massaro et al., 2019).

L076

L077 **Fig. 2** Results of the sensitivity analysis carried out on the Young's Modulus variations within each
L078 rock layer of the domain considering different configurations (stratified substratum model – nodes:
L079 4426; single magma chamber model – nodes: 4426; dual magma chamber model – nodes: 4161; dual
L080 magma chamber with conduits model – nodes: 3737). For each geological Unit (B, FC, GF, VD), the
L081 relative global variation in $L_2$ (%) is provided for $\sigma_1$ and $\sigma_3$. The $x(-)$ and $x(+)$ vectors indicate the
L082 Young's Modulus variation by an order of magnitude with respect to $x_{ref}$ vector, containing the stress
L083 values calculated by using the values of material's properties indicated in Table 2.

L084

L085 **Fig. 3** Spatial variation (%) of the $L_2$ norm's components at varying Young's Modulus for selected
L086 cases of Units B and VD: (a) Unit B in the stratified substratum model (nodes: 4426); (b) Unit B in
L087 the single magma chamber model (nodes: 4426); (c) Unit B in the dual magma chamber model

(nodes: 4161); (d) Unit VD in the dual magma chamber with conduits model (nodes: 3737). Symbols
$x(-)$ and $x(+)$ have the same meaning of Figure 2.

**Fig. 4** E-W gravitational modelling of the CVC domain. The scale of the mesh is expressed in Unit
of Design (1 UD = 1 km). The domain extends 60 km along the $x$-axis, and 30 km along the $z$-axis.
The number of nodes used in the mesh is set to 4426. The magnitude and pattern of the principal
stresses (dotted black lines) are reported for (a) the homogeneous stratigraphy (Unit FC =andesitic
lavas and pyroclastic deposits) and for (b) the not homogeneous stratigraphy (Unit FC; Unit B=
Cretaceous limestones and intrusive rocks forming the bed-rock underlying the CVC; Unit GF=
Quaternary alluvial, colluvial, and lacustrine deposits filling the graben; Unit VD= volcaniclastic
deposits covering the southern flank of the CVC). The blue line contours the unperturbed part of the
domain, which extends ca. 30 km horizontally and ca. 25 km vertically. Note that the scale of stress
values is the same for the all simulations.


**Fig. 5** E-W gravitational modelling of the CVC domain with a not homogeneous stratigraphy. The
magnitude and pattern of the principal stresses are reported for (a) the single magma chamber model
represented by a magma chamber ($2a = 14$ km and $2b = 3.6$ km) at 15 km of depth, and (b) the dual
magma chamber model composed of a 15 km-deep magma chamber ($2a = 14$ km and $2b = 3.6$ km)
and a shallow 6 km-deep one ($2a = 3.5$ km and $2b = 2$ km). The magma chambers are not connected.
The magmatic overpressures are set to 10 and 5 MPa for the 15 km-deep and 6 km-deep magma
chambers, respectively. The number of nodes is set to 4426 and 4161 for the single and dual magma
chamber models, respectively. Black dotted lines highlight the passage from different stress values.
The red dotted line in panel (b-i) indicates the formation of the stress arch. Note that the scale of
stress values are different for each panel in order to maximise the simulation details.

**Fig. 6** E-W gravitational modelling of the CVC domain with a not homogeneous stratigraphy
accounted for a dual magma chamber system connected by dykes via surface (deep magma chamber,
$2a = 14$ km and $2b = 3.6$ km at 15 km of depth; shallow magma chamber, $2a = 3.5$ km and $2b = 2$ km
at 6 km od depth). The magnitude and pattern of the principal stresses are shown. The number of
nodes used is set to 3737. The magmatic overpressures are set to 10 and 5 MPa for the 15 km-deep
and 6 km-deep magma chambers, respectively. The black dotted lines in panel (ii) highlight the
passage from different stress values. Note that the scale of stress values are different for each panel in
order to maximise the simulation details.

**Fig. 7** E-W gravitational modelling of the CVC domain with a not homogeneous stratigraphy

L124   considering the extensional field stress. The magnitude and pattern of the principal stresses are shown

L125   for the single magma chamber model (panels i-ii), the dual magma chamber model (panels iii-iv), the

L126   dual magma chamber with conduits model (panels v-vi-vii-viii). Note that in panel vii-viii the faults

L127   bordering the Colima graben are shown. For all configurations an extensive far-field stress of 5 MPa

L128   is applied at the lateral boundaries of the domain. In panels vii-viii the additional effect of the local

L129   extensive field is simulated using a reduced values of material's properties (Table 2). The magmatic

L130   overpressures are set to 10 and 5 MPa for the 15 km-deep and 6 km-deep magma chambers,

L131   respectively. Black dotted lines highlight the passage from different stress values. The red arrows

L132   indicate the direction of the applied far field stress. Note that the scale of stress values are different

L133   for each panel in order to maximise the simulation details.

L134

L135

L136

L137

L138

L139

L140

L141

L142   **Figures**

L143

L144

L145

L146

L147

L148

L149

L150

L151

L152    Figure 1

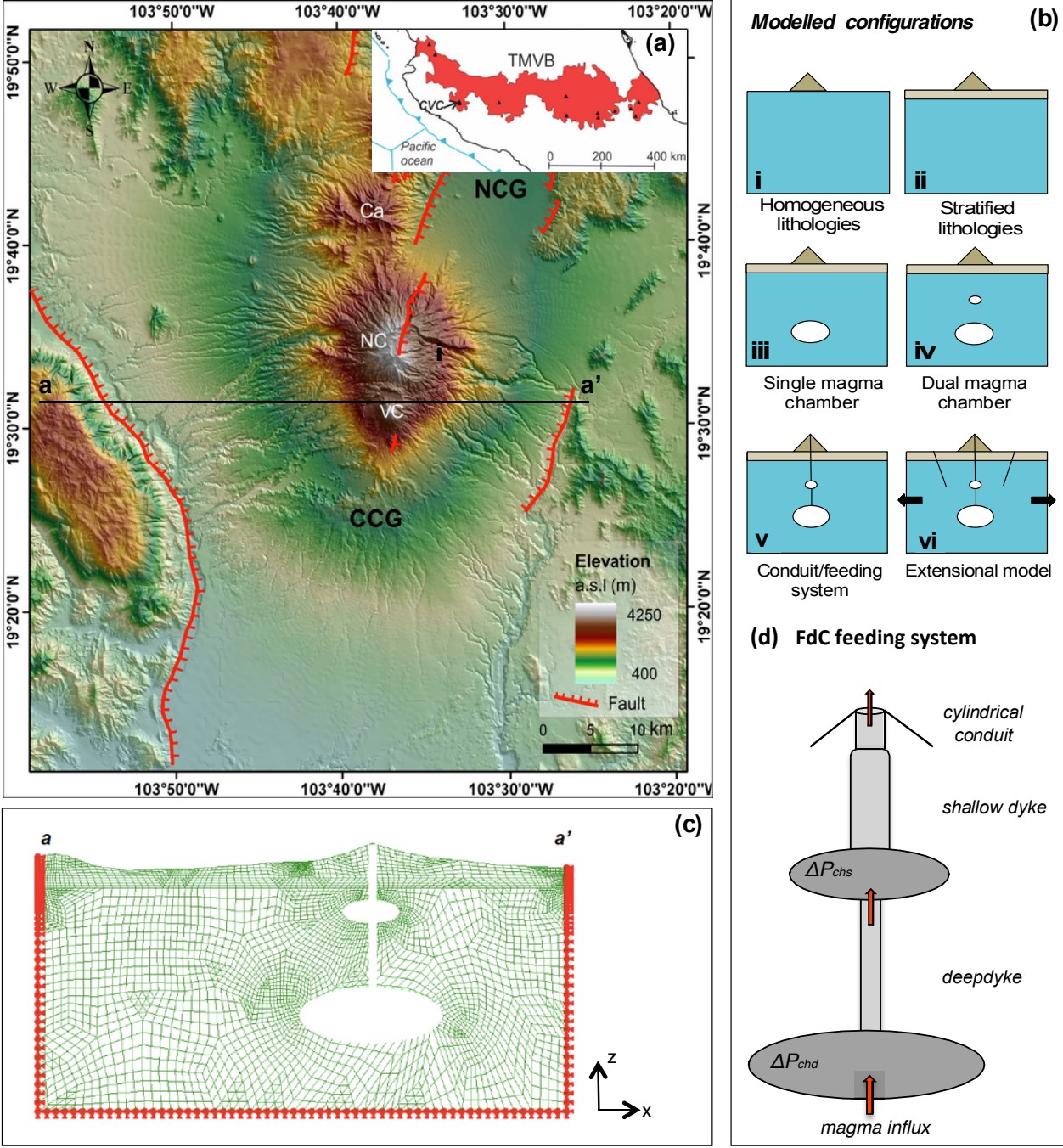

L153

L154

155    Figure 2

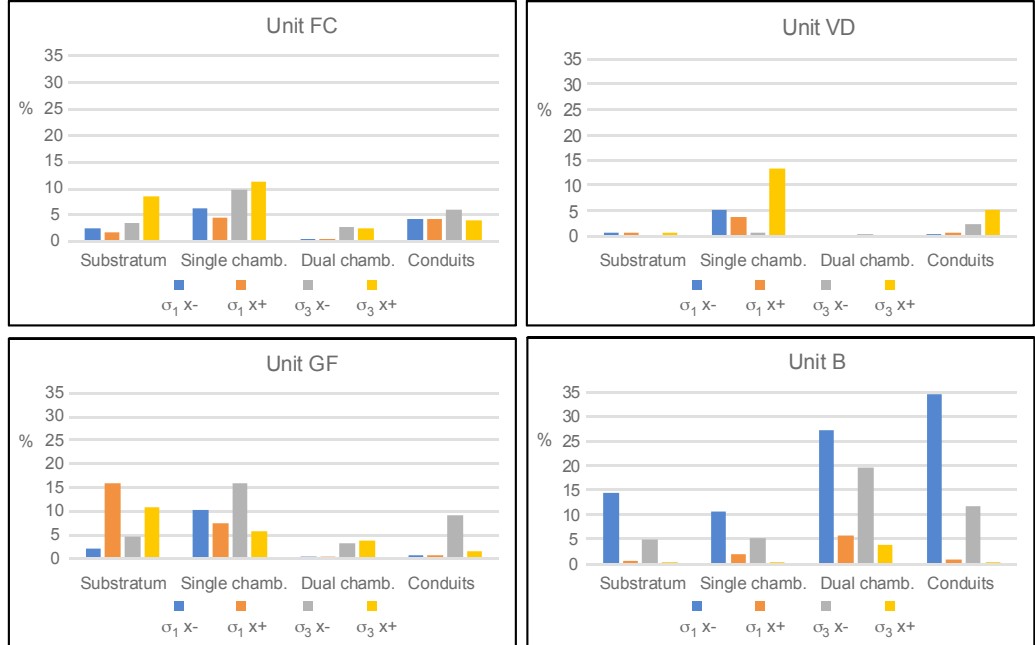

156

157

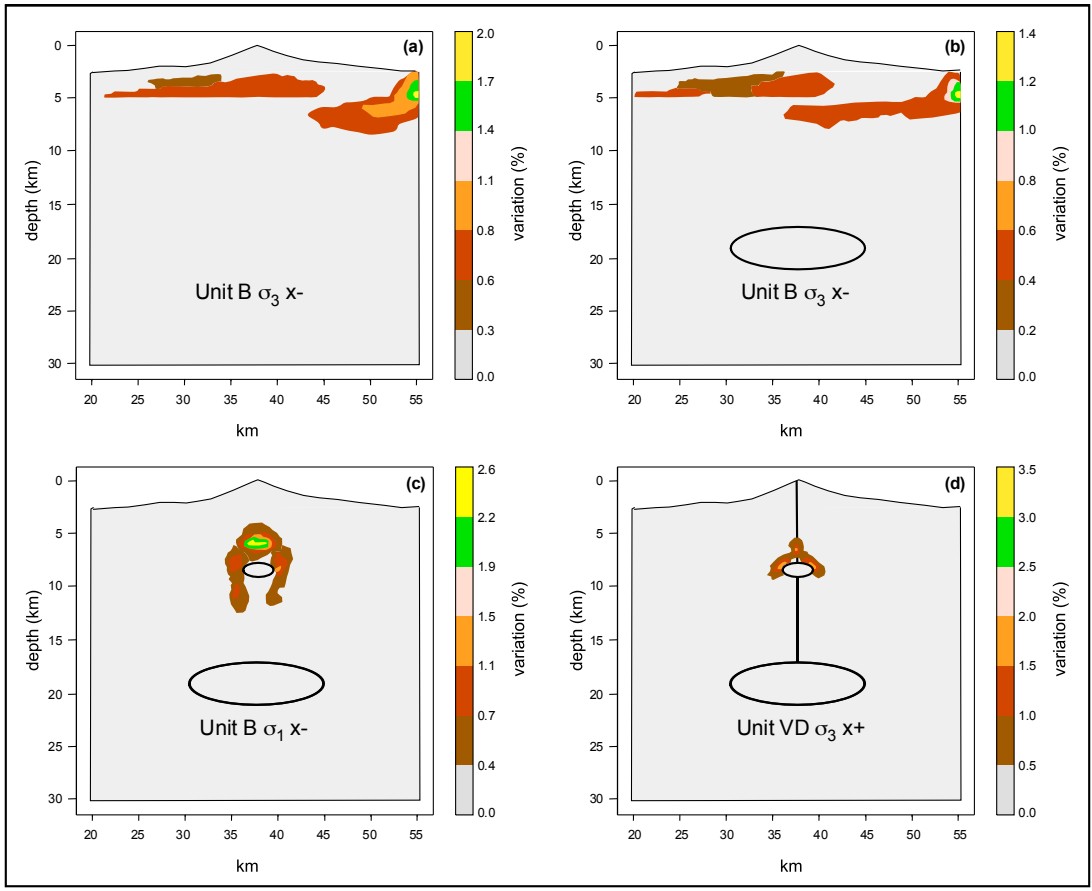

L161    Figure 4

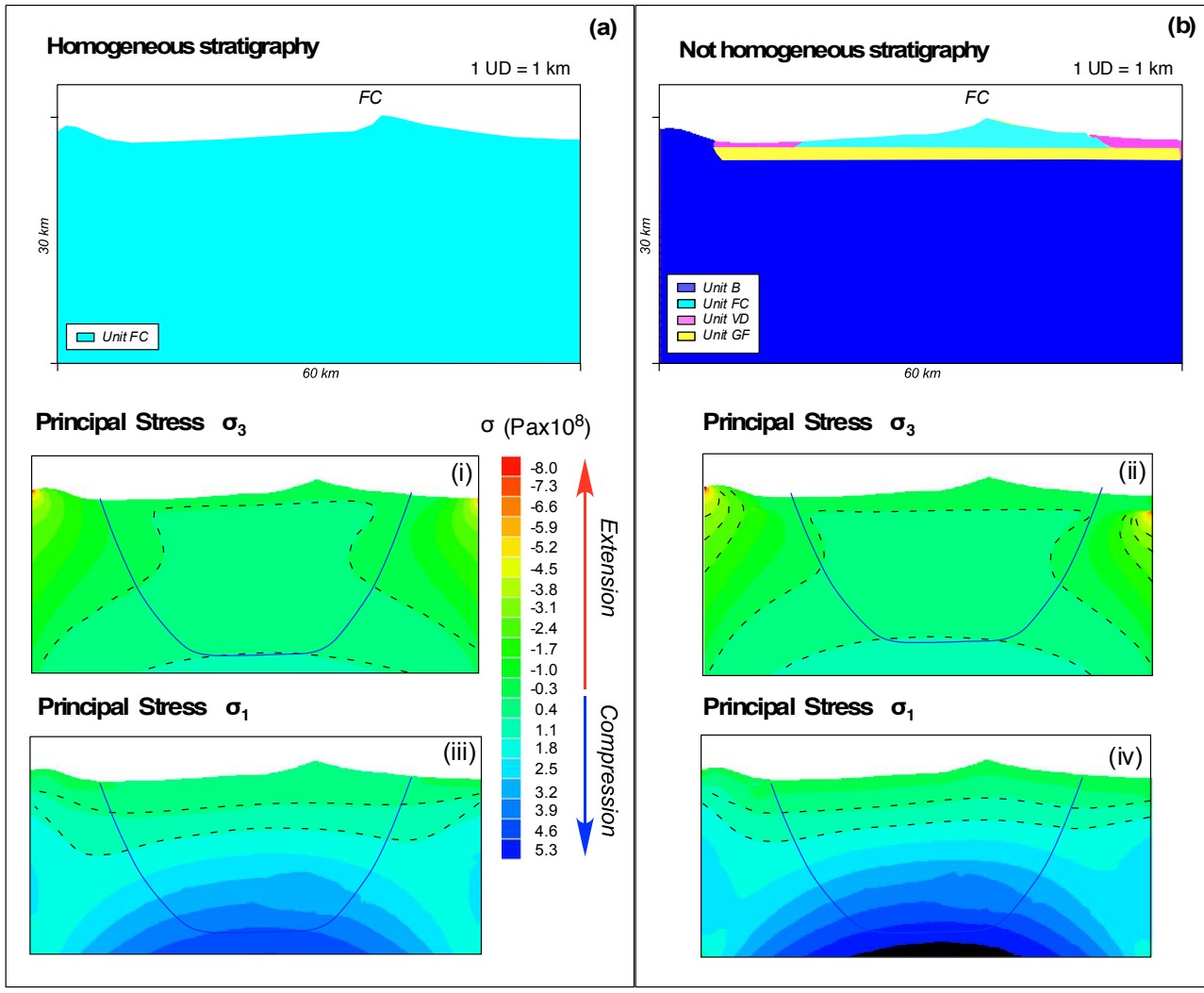

L162

L163

L164

L165

L166    Figure 5

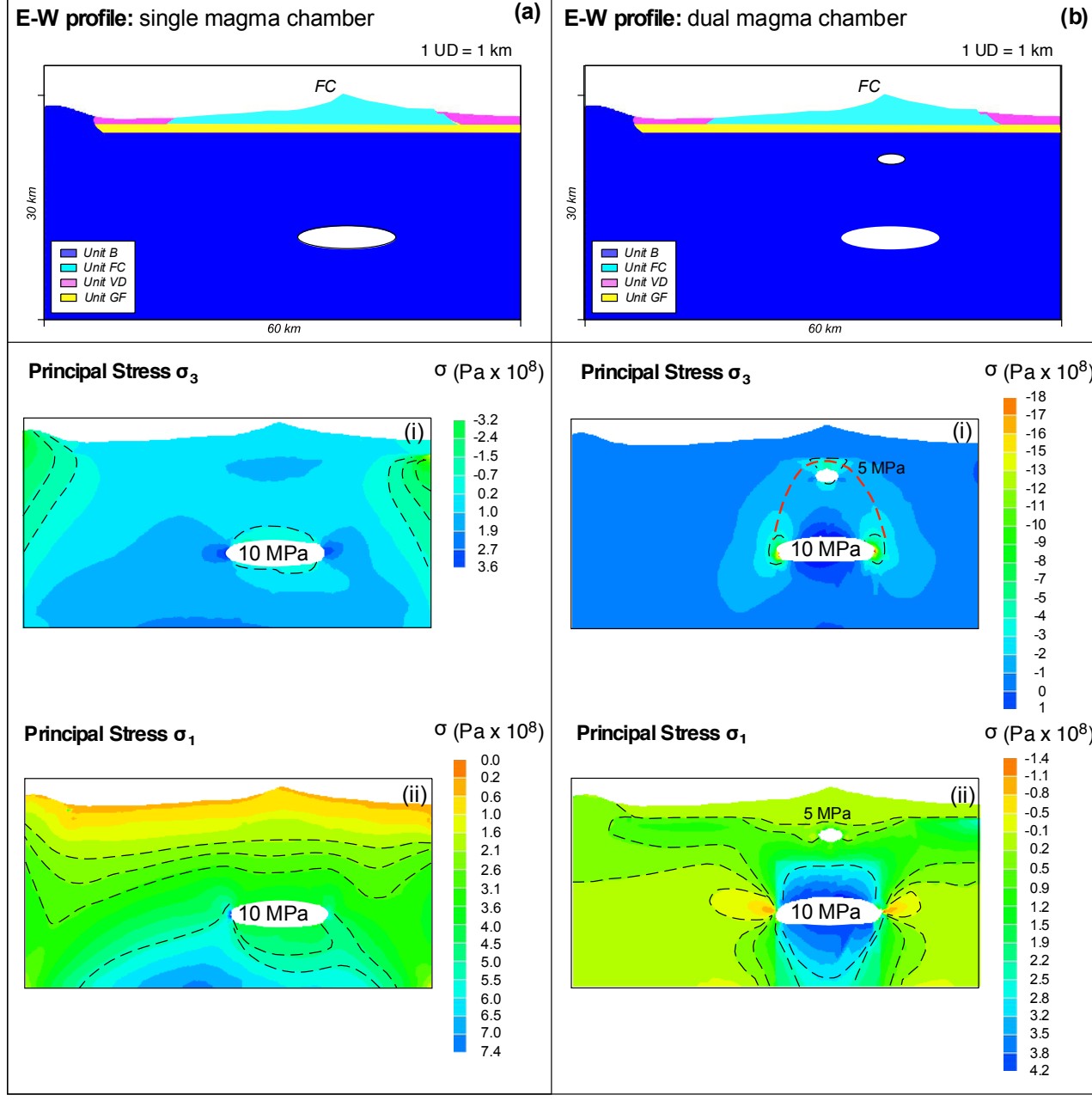

L167

L168

L169

L170

L171    Figure 6

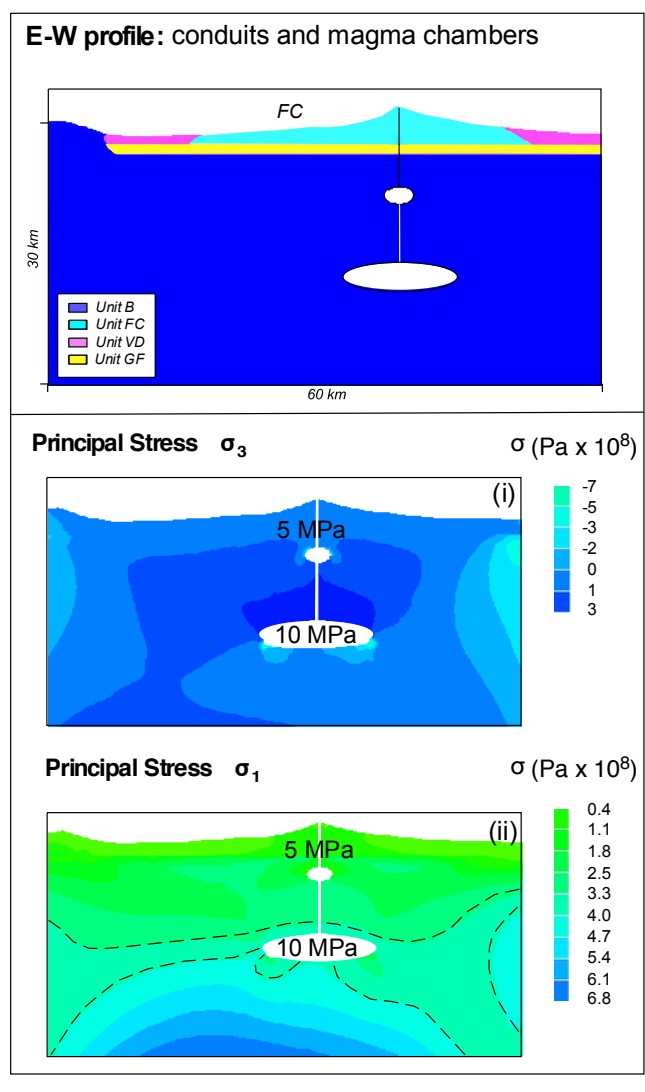

L172

L173

L174

L175

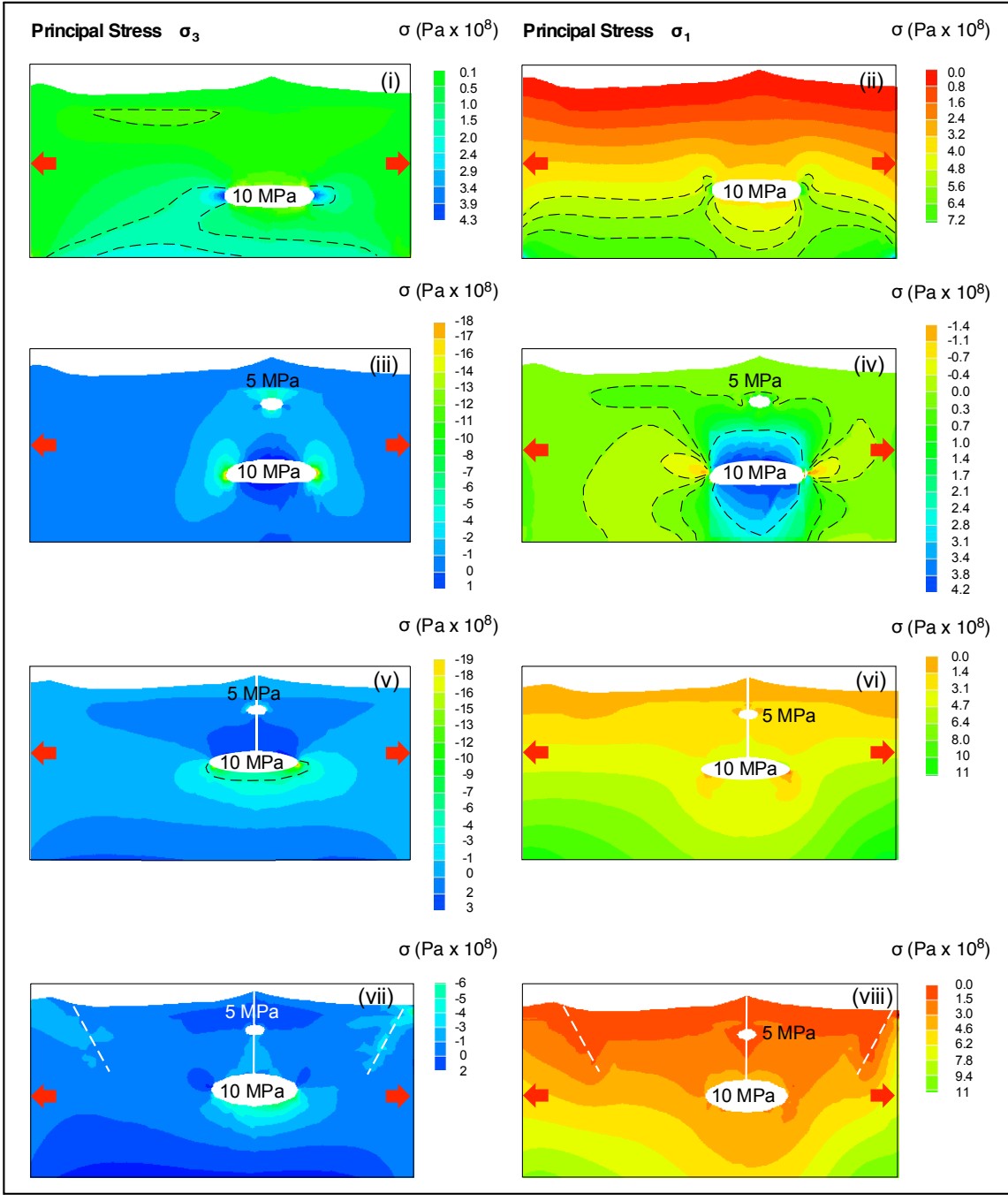

## Appendix 1

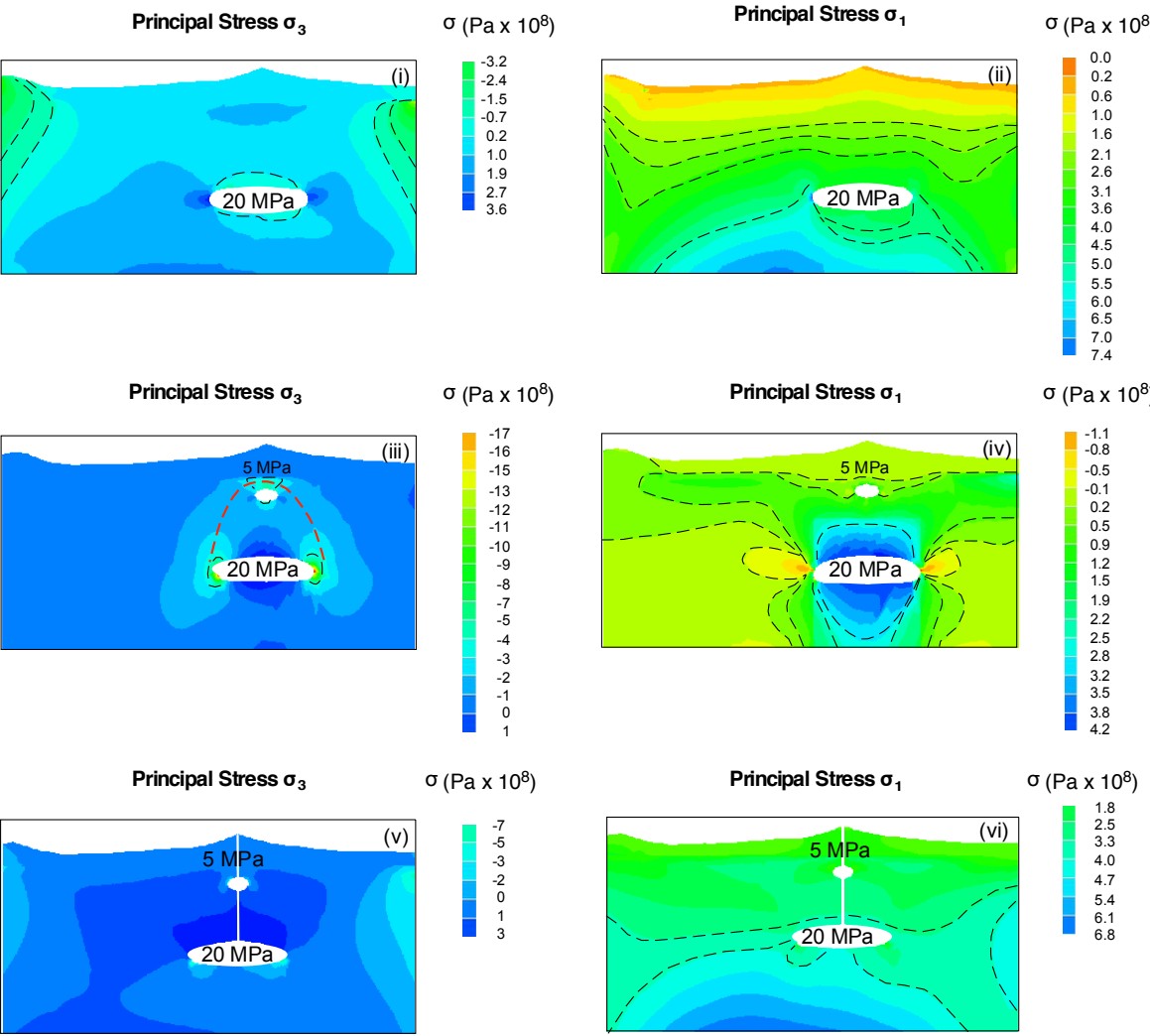

## Appendix 2

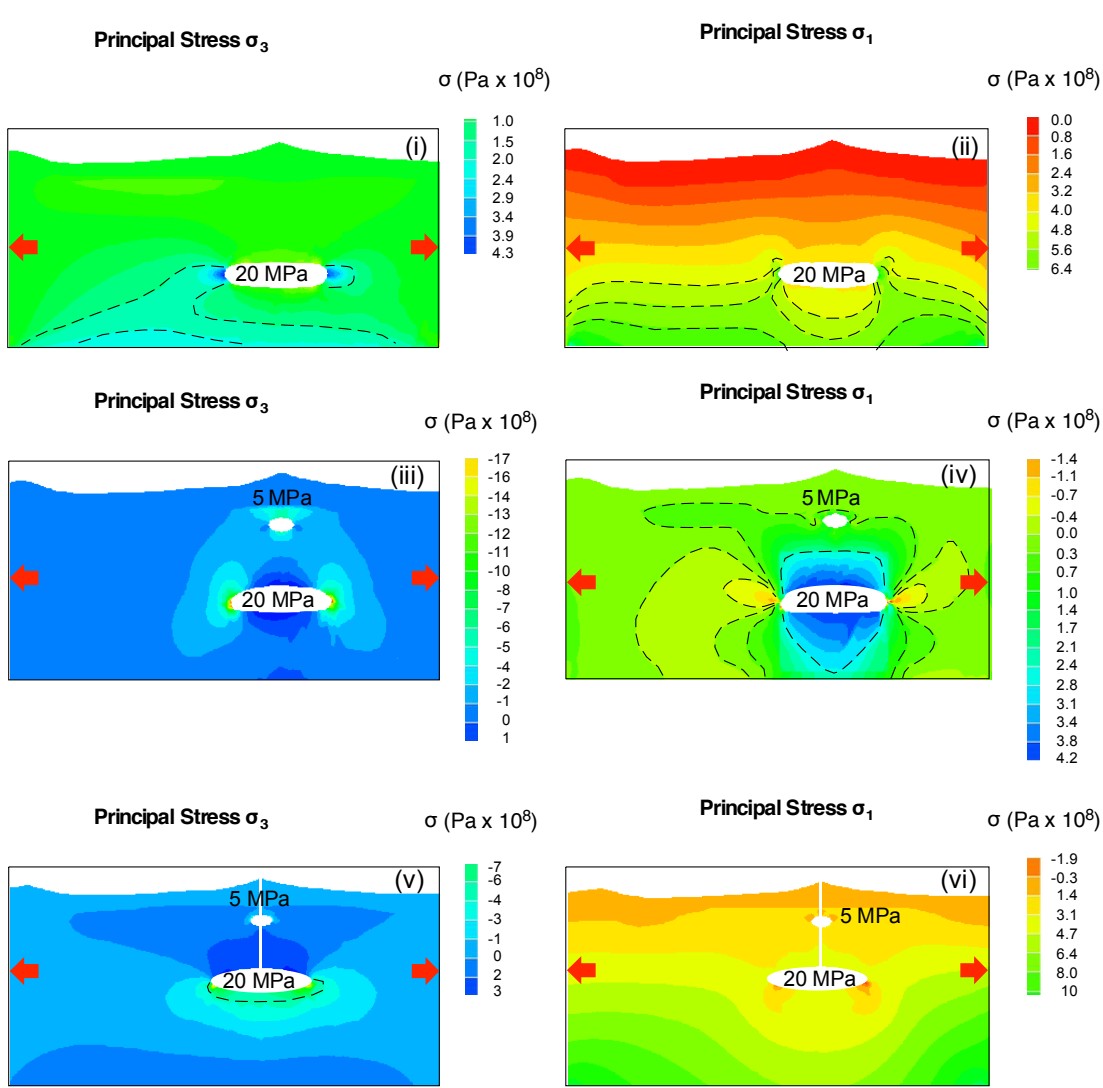