# Peer review of "Analysing stress field conditions of the Colima Volcanic Complex (Mexico) by integrating FEM simulations and geological data"

_Solid Earth, 2020_

## Referee Comment (RC1) · Virginie Pinel (Referee) · 1 Jul 2020

The manuscript entitled "Modelling stress field conditions of the Colima Volcanic Complex (Mexico)" by Massaro et al. investigates the added-value of integrating geological information in numerical models using the example of Colima Volcano, Mexico. The topic is certainly of high-interest but, in my opinion, the study suffers at this stage of several issues that should be addressed before considering publication. I will detail my main concerns below.

1. Main concerns:

[Figure]

* I enjoyed reading the introduction because the topic addressed is very important and this point is pretty well explained. But then I was somehow disappointed by the study itself. It is intended to evaluate the influence of geological data integrated into the modeling but only the stratigraphy and geometry of the plumbing system (see section 4) is considered. There is no real novelty in considering these aspects, which has been done in several studies (e.g. Cianetti et al, GJInt, 2012). I was expecting the authors to also take into consideration the existing faults after the long description of the structural/tectonic context of Colima volcano (fig1, section 2.1). In particular, the profile chosen for modeling cuts 2 faults of the Colima Rift, which are not considered in the models. Only a uniform extension applied at the lateral boundary is considered in some models. However, there are ways to consider a fault plane in 2D using a friction law (see Chaput et al, GRL 2014).

* Model assumptions are not clearly described.

-It is in 2D but it is not explained whether a plane strain or a plane stress approximation is considered, which is a key information. Usually models are performed in plane strain, which means that there is a stress component out of plane. . ..

-The way the gravitational loading is applied remains unclear. When applying body forces lithostatic stress field should also be applied but when a topography is considered, some iterations are required to find the initial state of stress consistent with both the topography, the rheology and the body forces as described in Chaput et al, GRL, 2014 or Cianetti et al, GJInt, 2012. Also with a lithostatic stress field, the load applied at the reservoir boundaries has to be a superposition of the overpressure and the lithostatic component. It is not explained in the manuscript. Also if a lithostatic stress field is applied both the minimum and maximum stress field should incease with depth. From the figures shown in the result section it is not the case for the minimum stress sigma3 and I don't really understand why.

* Illustrations should be improved to help the understanding. In some cases, the dimensions of the numerical box represented are not clearly reported, titles are unclear. I will detail later on each figure.

* Regarding the results and discussion of the Young modulus influence on the stress field, what matters are the ratios of the Young modulus considered in various layers and not the absolute value of the Young modulus in one given layer (I mean that if the Young modulus is mutliplied by 10 in each layer, no changes are expected except in the vicinity of the domain external boundaries). This fact is not clearly shown. Also I would recommend to cite the paper by Heap et al. published recently in the Journal of Volcanology and Geothermal Research (https://doi.org/10.1016/j.jvolgeores.2019.106684)

2. Minor points:

*Introduction: line 37, 41, before the chosen references list for numerical models I would put "e.g." because there are plenty of references that could be equally fairly cited here. Also I would add the reference to Cayol & Cornet, GRL, which is really a classical one. Line 57, I would suggest to also cite Albino et al., Geophysical journal international, 2010.

*Section 3.3 Line 213: it would be very helpful to show the mesh used. line 217, the boundary condition applied on the reservoir and dike walls should be explained. Line 229, the way gravity is expected to influence the failure condition is really depend on the rupture criterion considered (see for instance Albino, et al. JGR, 2018) Line 231, in Corbi et al, 2015, the trajectory of magma propagation is not influenced by gravity but by the deviatoric stress field induced by caldera unloading.

*section 4.2: Line 296 "During ascent to the surface, the dykes align themselves with the most energy-efficient orientation, which is roughly perpendicular to the least compressive principal stress axis $\sigma 3$ (e.g. Gonnermann and Taisne, 2015; Rivalta et al., 2019)." this is true providing the magma driving pressure remains small compared to the deviatoric stress (see Pinel et al, JGR, 2017 and Maccaferri et al. G3, 2019)

Section 5.1 Line 345, it would be important to show on a figure the reduced simulation domain selected for the sensibility analysis. Also for each unit, we would need to know the number of nodes considered (size of the vector space X). It is important because the larger variability of Unit B could be only due to the larger domain considered.

Figure 2 Figures labels and title should be improved, information of the number of nodes considered should be added.

Figure 3 : Figures labels and title should be improved, information of the number of node considered should be added. Limits of the different units should be shown. For panel A, it remains unclear to me which stress perturbation is considered as there is no reservoir.

Figure 4 : The topography doesn't look the same on each panel, which makes comparison difficult. No indication is provided on the orientation of the maximum and minimum compressive stress. I don't understand the term "distensive". Once again I don't understand why sigma3 does not increase with depth.

---

## Referee Comment (RC2) · Adelina Geyer (Referee) · 6 Jul 2020

Review of the paper: "Modelling stress field conditions of the Colima Volcanic Complex (Mexico) integrating FEM simulations and geological data" by S. Massaro and co-authors. The main objective of the present manuscript is to assess the influence of geological data on numerical simulations. As a case study, they investigate the stress field conditions occurring around the Colima Volcanic Complex (Mexico) using Finite Elements. In general, I find this paper very interesting, however I consider that some important points need to be addressed before publication.

General comments:

[Figure]

* When citing previous published works use "e.g." because the lists are not exhaustive and the cited articles are just a small example of the existing references.

* The introduction is a bit confusing to me because the title (and objectives) of the manuscript are focused on stress field and the first paragraph of the introduction is about volcano deformation. I would recommend the authors to rethink the introduction pointing out the importance of calculating the stress field in volcanic areas, which are the components of the stress field (i.e. those processes affecting/modifying it, etc). Then, they can connect all this with the FEM as a "numerical tool" to quantify/predict the stress field in a volcanic area.

* The objectives of the work are presented in two different parts of the introduction (L68-74 and 83-86). I suggest merging them at the end of the introduction.

* To better evaluate the influence of the diverse geological details on the results obtained, it would be more appropriate to carry out first a parametric study on the studied parameters (e.g., Young's modulus, Poisson ratio, magma chamber geometry). Keeping all parameters constant and changing one parameter at a time in a systematic way, is what really allows estimating (and quantifying) the influence of the individual parameters on the numerical results obtained (see, for example, Kinvig et al. 2009; Geyer & Gottsmann 2010). Once the parametric study has been done, results obtained can be applied to the case studies.

* The authors should be sure that all names mentioned in the text are included in the figures. For example, Figure 1 showing the geological setting of the CVC does not show the location of the Michoacan Block, the Chapala-Tula rift, etc.

* It would really help to include a sketch of the CVC plumbing system.

* The authors should show the mesh and also provide details about the size of the elements, not only the number.

*The authors should better describe how gravity is implemented and how is the resultant stress field derived from it considering the selected mechanical properties of the computational domain. Also, since there is topography, I do not understand what the authors mean with "Gravity in the host rock ( z <= 0)" . Is gravity not assigned for z values > 0? This part should be clearly explained because the "background stress field" generated by the gravitational loading may have a strong influence on the results obtained.

* Considering the size and depth of the deep magma chamber, I think that the domain boundaries are far too close to the area of study, specially to the W. This is also acknowledged by the authors (L382-389). Considering that the models are 2D (i.e. computational time is not too high compared to 3D models), it would have been safer to expand the limits of the computational domain further away from the magma reservoirs. The "displacement = 0 m" boundary condition has strong effects on the results obtained if the boundary is too close to the pressure source.

L49-51: I do not understand this sentence. What do the authors mean by "boundary representation"? Please, we sure that you are not confusing the Boundary Element Method (BEM) with the Finite Element Method (FEM).

L55-58: I would mention also the use of FEM for fluid dynamics or thermal problems to illustrate their application to solve other type of physical equations, not only those related to rock mechanics (e.g., Bea 2010; Gutiérrez & Parada 2010; Gelman et al. 2013; Douglas et al. 2016).

L59-60: Use "e.g."

L60-62: Add references and indicate in GPa what is meant by "stiff" and "low".

L67-69: Include some references to illustrate what kind of publications already exist.

L74: I think something is missing in this sentence.

L81-83: Please, revise this sentence. I think that something is not correct in the English, a native English speaker should verify it.

L87-92: What overpressure? This sentence is confusing. All this paragraph should come much earlier in the introduction, when presenting the problematic the authors want to solve. If the idea is to highlight the limitations of the elastic approach used in the models, this section should be move to the "Methodology" section.

L95: The CVC acronym has been already explained.

L112: Where is all this information shown in Figure 1?

L130: "a.s.l."

L186-188: What do the authors mean with "complex" structure?

L193: Extension or extent?

L198: Indicate the website and what INEGI means.

L215: Which geological units? The magma chamber? The rock layers? This sentence is confusing.

L222-224: Since the authors have already extensively described it in the previous section maybe they should refer to their own text (and figure) here.

L224-227: Not sure which is the objective of this sentence, as the authors do not explain the overpressure assigned to their models in this paragraph. Is something missing?

L228: Commas are missing after between and with, otherwise the sentence is difficult to understand.

L258: Please, add references.

L260-261: Please, add references.

Figures:

Figure 1: Indicate the north arrow in (a)

Figure 4: The color different between Unit VD and GF is practically undistinguishable. It seems that the top-left image has a different orientation than the others. The selected color scale is strongly conditioned by the boundary effects at the right and left corners at the free surface. The authors should recalibrate the color scale so that the gravity stress field is visible also at shallower depths. Now is all in green.

Figure 5: It is really confusing to have to color scales for (a) and (b). It is difficult to compare the results between both models and the effect of the shallow reservoir. Has model b the gravitational loading implemented? It is strange to me to see that model provides negative sigma 1 values at such depths (i.e. 15 km).

Figure 6: I strongly recommend using another color scale, similar to the one in Figure 5 going from red to blue colors. In the sigma 1 picture many details are lost because of it.

Figure 7: Same comment as in Figure 5. Is in the model in the middle gravity implemented? To facilitate the comparison among all pictures, the same color scale for all sigma 1 and for all sigma 3 should be assigned. Otherwise is very confusing because the same colors are sometimes <0 and other times >0.

---

## Editor Comment (EC1) · Joan Marti (Editor) · 7 Jul 2020

We have not received two reviews of your mansucript. Both referees have done a good job and provided very detailed reviews. Please comment on them, point by point, and prepare a revised version of the manuscript according to these comments. The review will proceed after that.
* * *

---

## Author Comment (AC1) · 11 Aug 2020

**Author's response**

**Reviewer#1 – Virginie Pinel**

1. Main points

1) I enjoyed reading the introduction because the topic addressed is very important and this point is pretty well explained. But then I was somehow disappointed by the study itself. It is intended to evaluate the influence of geological data integrated into the modeling but only the stratigraphy and geometry of the plumbing system (see section 4) is considered. There is no real novelty in considering these aspects, which has been done in several studies (e.g. Cianetti et al, GJInt, 2012). I was expecting the authors to also take into consideration the existing faults after the long description of the structural/tectonic context of Colima volcano (fig1, section 2.1). In particular, the profile chosen for modeling cuts 2 faults of the Colima Rift, which are not considered in the models. Only a uniform extension applied at the lateral boundary is considered in some models. However, there are ways to consider a fault plane in 2D using a friction law (see Chaput et al, GRL 2014).

*We thank the reviewer for the useful comments. The manuscript has been submitted for the special issue on the use of geological data for constraining numerical models. We focused our attention to geology (stratigraphic data, geometry of the feeding system) of the Colima Volcano area. However, we are aware the approach is not new, but it is new for Colima volcano. Moreover, it provides further insights on the limitations of FEM analysis when some geological features are unknown or not considered. The insertion of a constant extensive stress is not trivial, because it simulates the stress state that dominates the Colima rift over long times, affecting the computational domain. However, we agree that the fault movement (and the fault plane itself) may induce different perturbation in the stress state of the conduit feeding system area. However, although LISA cannot include a frictional law as in Chaput et al. (2014) and cannot provide the solution for two separated domains at the same time, in order to consider Reviewer#1's criticism, in the revised version we considered an approximate description of the effects of faults. In particular, since mature fault zones are generally made of complex highly deformed and low cohesive materials where most of the fault displacement is accommodated and surrounded by a fractured damage zone, in this revised version of the manuscript we have provided a new final configuration considering the faults of the Colima graben as "damage zones". Technically, we have selected two bands of elements by assigning more degraded elastic properties (as reported in Jeanne et al. 2017). Results with considering the fault are provided in the new Figure 7.*

2) Model assumptions are not clearly described.
- It is in 2D but it is not explained whether a plane strain or a plane stress approximation is considered, which is a key information. Usually models are performed in plane strain, which means that there is a stress component out of plane.
*We thank the Reviewer for such a point. Now we clearly stated that it is a plane strain in the text.*

-The way the gravitational loading is applied remains unclear. When applying body forces lithostatic stress field should also be applied but when a topography is considered, some iterations

are required to find the initial state of stress consistent with both the topography, the rheology and the body forces as described in Chaput et al, GRL, 2014 or Cianetti et al, GJInt, 2012. Also with a lithostatic stress field, the load applied at the reservoir boundaries has to be a superposition of the overpressure and the lithostatic component. It is not explained in the manuscript. Also if a lithostatic stress field is applied both the minimum and maximum stress field should increase with depth. From the figures shown in the result section it is not the case for the minimum stress sigma3 and I don't really understand why.

*We described better this point in the text. The load applied to magma chambers is exactly the same requested by the reviewer (a superposition of the magmatic overpressure and the lithostatic component). Both $\sigma_1$ and $\sigma_3$ increase with depth, as shown in Figure 4 where it can be seen $\sigma_3$ passes from zero to 110 MPa at depth. $\sigma_1$ passes from zero to ca 390 MPa. Probably, the reviewer was erroneously confused by the same colour scale applied to both Figures. In order to avoid confusion we clearly highlighted this point about colour scale in the revised text.*

3) Illustrations should be improved to help the understanding. In some cases, the dimensions of the numerical box represented are not clearly reported, titles are unclear. I will detail later on each figure. *Figures have been improved in this revised version.*

2. Minor points

1) Regarding the results and discussion of the Young modulus influence on the stress field, what matters are the ratios of the Young modulus considered in various layers and not the absolute value of the Young modulus in one given layer (I mean that if the Young modulus is mutliplied by 10 in each layer, no changes are expected except in the vicinity of the domain external boundaries). This fact is not clearly shown. Also I would recommend to cite the paper by Heap et al. published recently in the Journal of Volcanology and Geothermal Research (https://doi.org/10.1016/j.jvolgeores.2019.106684).

*In the light of what reported in Heap et al. (2020) about the selection of the most appropriate Young's Modulus in modelling, we would like to point out that in our parametric study we aimed to show what influence has the choice of different values of Young's Modulus in each geological unit at changing different feeding system configurations (i.e. single chamber, double chambers not connected and connected with conduits). The results demonstrated that the changes are not only in the vicinity of the external boundaries as expected, but also (even small) around the shallow magma chamber and conduits.*

2) Introduction: line 37, 41, before the chosen references list for numerical models I would put "e.g." because there are plenty of references that could be equally fairly cited here. Also I would add the reference to Cayol & Cornet, GRL, which is really a classical one. Line 57, I would suggest to also cite Albino et al., Geophysical journal international, 2010. *Added.*

3) Section 3.3 Line 213: it would be very helpful to show the mesh used. *Added in Figure 1 (panel c);* line 217, the boundary condition applied on the reservoir and dike walls should be explained. *We explained in the text.*

Line 229, the way gravity is expected to influence the failure condition is really depend on the rupture criterion considered (see for instance Albino, et al. JGR, 2018). *We specified this adding the suggested reference.*

Line 231, in Corbi et al, 2015, the trajectory of magma propagation is not influenced by gravity but by the deviatoric stress field induced by caldera unloading. *Thanks for such a comment. We corrected it in the text removing the reference.*

4) Section 4.2: Line 296 "During ascent to the surface, the dykes align themselves with the most energy-efficient orientation, which is roughly perpendicular to the least compressive principal stress axis σ3 (e.g. Gonnermann and Taisne, 2015; Rivalta et al., 2019)." this is true providing the magma driving pressure remains small compared to the deviatoric stress (see Pinel et al, JGR, 2017 and Maccaferri et al. G3, 2019). *We specified this in the text, also adding the suggested references.*

5) Section 5.1 Line 345, it would be important to show on a figure the reduced simulation domain selected for the sensibility analysis. Also for each unit, we would need to know the number of nodes considered (size of the vector space X). It is important because the larger variability of Unit B could be only due to the larger domain considered.
*We added the number of nodes for each geological Unit, in Table 1 (beside the number of elements).*

6) Figure 2: Figures labels and title should be improved, information of the number of nodes considered should be added. *We added the requested information in the captions.*

7) Figure 3: Figures labels and title should be improved, information of the number of node considered should be added. Limits of the different units should be shown. For panel A, it remains unclear to me which stress perturbation is considered as there is no reservoir.

*We improved the labels of Figures 2-3 adding the requested information. We preferred to not indicate the limits of the different units because they would not be clear with this scale. For the panel (a) of Figure 3, the stress perturbation is only due to the lithostatic load.*

8) Figure 5: The topography doesn't look the same on each panel, which makes comparison difficult. No indication is provided on the orientation of the maximum and minimum compressive stress. I don't understand the term "distensive". Once again I don't understand why sigma3 does not increase with depth.
*Topography was corrected. The term "distensive" was changed into "extension". We do not understand to what refers the reviewer when talking about orientation of maximum and minimum compressive stress. We already replied earlier about the increase of σ3 with depth.*

**Reviewer#2 – Adelina Geyer**

1) General points

1) When citing previous published works use "e.g." because the lists are not exhaustive and the

cited articles are just a small example of the existing references. *We added "e.g." in the lists where citations are not exhaustive.*

2) The introduction is a bit confusing to me because the title (and objectives) of the manuscript are focused on stress field and the first paragraph of the introduction is about volcano deformation. I would recommend the authors to rethink the introduction pointing out the importance of calculating the stress field in volcanic areas, which are the components of the stress field (i.e. those processes affecting/modifying it, etc). Then, they can connect all this with the FEM as a "numerical tool" to quantify/predict the stress field in a volcanic area.

*We agree to point out the issue of calculating the stress field in the volcanic areas within the introduction. We added this part in the Introduction.*

3) The objectives of the work are presented in two different parts of the introduction (L68-74 and 83-86). I suggest merging them at the end of the introduction. *We merged the sentences at the end of the Introduction.*

4) To better evaluate the influence of the diverse geological details on the results obtained, it would be more appropriate to carry out first a parametric study on the studied parameters (e.g., Young's modulus, Poisson ratio, magma chamber geometry). Keeping all parameters constant and changing one parameter at a time in a systematic way, is what really allows estimating (and quantifying) the influence of the individual parameters on the numerical results obtained (see, for example, Kinvig et al. 2009; Geyer & Gottsmann 2010). Once the parametric study has been done, results obtained can be applied to the case studies.

*The parametric study was carried out on Young's modulus but may be easily carried out for Poisson's ratio. In any case, the investigation of different parameters (i.e. rock mechanics and magma chamber geometry) would result in increasing the length of the paper, which is instead focused on influence of geological data (i.e. stratigraphy of the domain, feeding system geometry, effect of the rifting) on FEM simulations. However, we cited the papers mentioned by the reviewer in order to highlight the importance of the rock mechanics parameters.*

5) The authors should be sure that all names mentioned in the text are included in the figures. For example, Figure 1 showing the geological setting of the CVC does not show the location of the Michoacan Block, the Chapala-Tula rift, etc.

*We changed the citation in the text considering what reported in Norini et al. (2010-Figure 1) in which all mentioned geological structures are represented.*

6) It would really help to include a sketch of the CVC plumbing system. *We added it in the new version of Figure 1 (panel d).*

7) The authors should show the mesh and also provide details about the size of the elements, not only the number. *We included the mesh used in our modelling in Fig. 1 – panel (c), and we provided more information in the text about the mesh elements of each geological unit.*

8) The authors should better describe how gravity is implemented and how is the resultant stress

field derived from it considering the selected mechanical properties of the computational domain. Also, since there is topography, I do not understand what the authors mean with "Gravity in the host rock (z <= 0)". Is gravity not assigned for z values > 0? This part should be clearly explained because the "background stress field" generated by the gravitational loading may have a strong influence on the results obtained.

*We provided a better explanation in the text. We also removed "( z <= 0)".*

9) Considering the size and depth of the deep magma chamber, I think that the domain boundaries are far too close to the area of study, specially to the W. This is also acknowledged by the authors (L382-389). Considering that the models are 2D (i.e. computational time is not too high compared to 3D models), it would have been safer to expand the limits of the computational domain further away from the magma reservoirs. The "displacement = 0 m" boundary condition has strong effects on the results obtained if the boundary is too close to the pressure source.

*We showed the boundary effect in Figure 4. It is evident how the area comprising the feeding system (visible in Figs. 5-6-7) is not affected by any false result. To enlarge the domain would result in degrading of the details of the FEM simulation, which are already biased by the huge area at present considered for simulations.*

10) L49-51: I do not understand this sentence. What do the authors mean by "boundary representation"? Please, be sure that you are not confusing the Boundary Element Method (BEM) with the Finite Element Method (FEM).

*We revised the paragraph in the text in order to not be confusing. In particular, beside FEM, we aimed to describe another common way to describe the geological units by using BEM.*

11) L55-58: I would mention also the use of FEM for fluid dynamics or thermal problems to illustrate their application to solve other type of physical equations, not only those related to rock mechanics (e.g., Bea 2010; Gutiérrez & Parada 2010; Gelman et al. 2013; Douglas et al. 2016). *Done.*

12) L59-60: Use "e.g." *Done.*

13) L60-62: Add references and indicate in GPa what is meant by "stiff" and "low". *Done.*

14) L67-69: Include some references to illustrate what kind of publications already exist. *Done.*

15) L74: I think something is missing in this sentence.

16) L81-83: Please, revise this sentence. I think that something is not correct in the English, a native English speaker should verify it. *Corrected in the text.*

17) L87-92: What overpressure? This sentence is confusing. All this paragraph should come much earlier in the introduction, when presenting the problematic the authors want to solve. If the idea is to highlight the limitations of the elastic approach used in the models, this section should be move to the "Methodology" section. *We specified: "estimate of magmatic overpressure".*

18) L95: The CVC acronym has been already explained. L112: Where is all this information shown in Figure 1? L130: "a.s.l."L186-188: What do the authors mean with "complex" structure? L193: Extension or extent?

*Corrections made in the text. We changed the citation from Figure 1 to Norini et al. 2010 – Fig 1). For "complex structure" we referred to the dual magma chamber system.*

19) L198: Indicate the website and what INEGI means. *Done.*

20) L215: Which geological units? The magma chamber? The rock layers? This sentence is confusing. *We referred to the extent of the rock layers, described in the following text and detailed in Table 2.*

21) L222-224: Since the authors have already extensively described it in the previous section maybe they should refer to their own text (and figure) here.

*In the previous section (3.3 Modelling approach) we referred to Spica et al. 2017 but other parameters used in our modeling are described in other papers (i.e., Massaro et al. 2018, 2019) therefore we think it is useful add here these citations. We also added the reference to Figure 1d.*

22) L224-227: Not sure which is the objective of this sentence, as the authors do not explain the overpressure assigned to their models in this paragraph. Is something missing? *No, in this sentence we only reported a general statement.*

23) L228: Commas are missing after between and with, otherwise the sentence is difficult to understand. *Done.*

24) L258: Please, add references. L260-261: Please, add references. Figures: Figure 1: Indicate the north arrow in (a). *Done for references. Figure 1: the North is on the top, left-side corner.*

25) Figure 4: The color different between Unit VD and GF is practically undistinguishable. It seems that the top-left image has a different orientation than the others. The selected color scale is strongly conditioned by the boundary effects at the right and left corners at the free surface. The authors should recalibrate the color scale so that the gravity stress field is visible also at shallower depths. Now is all in green.

*We changed the colour of Unit GF. About the colour scales, they were set in a way they represent all the four panels, in order to facilitate comparison. We are aware of the similarity of green colors, and for this we separated the different colours with dashed line to indicate changes in the stress value.*

26) Figure 5: It is really confusing to have to color scales for (a) and (b). It is difficult to compare the results between both models and the effect of the shallow reservoir. Has model b the gravitational loading implemented? It is strange to me to see that model provides negative sigma 1 values at such depths (i.e. 15 km).

*The different colour scales were used, in this case, just for avoiding the problem highlighted in the*

*previous point by the reviewer. To have a common scale would result in too large stress classes (with the same colour) that would prevent the readability of each example. Both models have the gravitational loading implemented. You have to bear in mind that changing geological conditions results in changes in while stress in the simulation, which prevents the use of a common colour scale in LISA. The moderate negative $\sigma_1$ values are due to the effects of magma chamber overpressure with respect to the lithostatic load.*

27) Figure 6: I strongly recommend using another color scale, similar to the one in Figure 5 going from red to blue colors. In the sigma 1 picture many details are lost because of it.

*It is not possible to freely set the colour scale in LISA. The alternative colour scales provided by LISA are grey-scale and red-blue but they do not provide a better visualization than this shown in Figures (rainbow colour scale). Unfortunately, the details of $\sigma_1$ are lost also in this case.*

28) Figure 7: Same comment as in Figure 5. Is in the model in the middle gravity implemented? To facilitate the comparison among all pictures, the same color scale for all sigma 1 and for all sigma 3 should be assigned. Otherwise is very confusing because the same colors are sometimes <0 and other times >0.

*Also in this case the gravitational loading has been implemented. As already stated before, the addition of different geological details changes the stress distribution and its value. For this, it is not possible to use the same colour scale for all the simulations, otherwise we would have very broad, poorly informative scale of stress values.*

On behalf of the authors
Sincerely,

*Silvia Massaro*

---

## Author Comment (AC2) · 11 Aug 2020

Dear Referee,

we thank you for the revision made on our manuscript. Please find attached our response (see autors_response.pdf file as supplment).

Best Regards, Silvia Massaro

Please also note the supplement to this comment:
https://se.copernicus.org/preprints/se-2020-82/se-2020-82-AC2-supplement.pdf

---

## Author Comment (AC3) · 11 Aug 2020

Dear Editor,

please find attached in the interactive discussion the revised version of our manuscript (file in pdf with the marked changes) along with the author's response (pdf) including, point by point, the requested information asked by the referees. In the following, we would like to convert the manuscript in LaTeX version. Best Regards, Silvia Massaro

---

## Author Response (AR2)

Dear Executive Editor SE, we are deeply grateful for the critical comments that helped us to improve the clarity and quality of the manuscript. In this new version we have revised the English writing, shortened and reorganised the text in order to avoid repetitions and too generic statements. We accepted all the corrections listed in your comment.

In particular, we defined the magmatic pressure as either excess pressure ($\Delta P_e$, magmatic minus lithostatic pressure but below the tensile strength of wall rocks) or over pressure (or driving pressure $\Delta P_o$, which is the magmatic pressure exceeding tensile strength of wall rocks) according to Gudmundsson (2012). The first pertains to the FEMs using isolated magma chambers (single or double), while the second is used for models with connected magma chambers (with conduit/feeding system).

Please find attached here the manuscript with the tracked changes, along with a clean version. We hope this presentation can fulfil your requests.

With our best regards,

On the behalf of Authors

Silvia Massaro

[revised manuscript text omitted]

* * *
Margin comments (tracked changes):

Silvia 16/9/20 21:23

Silvia 16/9/20 21:23

Utente di Microsoft Office 22/9/20 16:18

Silvia 16/9/20 21:23

Silvia 16/9/20 21:23

Silvia 16/9/20 21:23

Antonio Costa 27/9/20 12:08

Silvia 16/9/20 21:29

Silvia 16/9/20 21:33

Silvia 16/9/20 21:33

Utente di Microsoft Office 22/9/20 16:21

Utente di Microsoft Office 22/9/20 16:21

Utente di Microsoft Office 22/9/20 16:21

Silvia 16/9/20 18:00

Silvia 16/9/20 17:54

Silvia 16/9/20 18:20

Utente di Microsoft Office 22/9/20 16:22

Silvia 19/9/20 17:36

Silvia 19/9/20 17:36

[revised manuscript text omitted]

Silvia 16/9/20 22:15

Utente di Microsoft Office 22/9/20 18:50

Silvia 19/9/20 19:23

Utente di Microsoft Office 25/9/20 11:44

Silvia 28/9/20 09:59

Silvia 28/9/20 09:59

Silvia 28/9/20 09:59

Silvia 28/9/20 10:00

Silvia 28/9/20 09:59

Utente di Microsoft Office 25/9/20 11:49

Antonio Costa 27/9/20 12:24

Silvia 28/9/20 10:03

Utente di Microsoft Office 22/9/20 19:55

Silvia 19/9/20 19:25

Utente di Microsoft Office 22/9/20 19:55

Silvia 19/9/20 19:25

Utente di Microsoft Office 22/9/20 18:57

Utente di Microsoft Office 22/9/20 21:58

Silvia 25/9/20 14:19

Antonio Costa 27/9/20 12:27

Silvia 25/9/20 14:20

[revised manuscript text omitted]

**5 Results**

Silvia 19/9/20 19:45

Utente di Microsoft Office 22/9/20 20:21

Silvia 19/9/20 19:52

Silvia 16/9/20 19:14

Utente di Microsoft Office 22/9/20 20:06

Silvia 19/9/20 19:56

Utente di Microsoft Office 22/9/20 20:13

Silvia 19/9/20 19:57

Utente di Microsoft Office 25/9/20 11:58

Silvia 19/9/20 19:58

Utente di Microsoft Office 22/9/20 20:14

Silvia 19/9/20 19:59

Utente di Microsoft Office 22/9/20 21:57

In this section we reported the sensitivity analysis carried out to quantify the approximation of the Young Modulus variation on FEM outputs, and the description of the model outputs when adding complexity to the input geological/geophysical data.

*5.1 Sensitivity analysis of Young Modulus*

[revised manuscript text omitted]

Spostato in su [4]: In Figure 2 are ... [67]
Utente di Microsoft Office 22/9/20 22:24
Silvia 19/9/20 20:21
Utente di Microsoft Office 22/9/20 22:25
Silvia 19/9/20 20:22
Silvia 20/9/20 00:23
Utente di Microsoft Office 22/9/20 22:27
Silvia 20/9/20 00:34
Utente di Microsoft Office 22/9/20 22:27
Silvia 20/9/20 00:34
Utente di Microsoft Office 22/9/20 22:28
Silvia 20/9/20 00:37
Silvia 20/9/20 00:39
Utente di Microsoft Office 22/9/20 22:32
Silvia 20/9/20 00:38
Silvia 20/9/20 00:41
Antonio Costa 27/9/20 12:47
Silvia 20/9/20 00:41
Utente di Microsoft Office 22/9/20 22:38
Silvia 20/9/20 00:41
Utente di Microsoft Office 22/9/20 22:38
Silvia 20/9/20 00:42
Utente di Microsoft Office 22/9/20 22:38
Silvia 20/9/20 00:43
Silvia 20/9/20 00:43
Silvia 20/9/20 00:45
Utente di Microsoft Office 22/9/20 22:40
Silvia 20/9/20 00:44
... [83]
Utente di Microsoft Office 22/9/20 22:40
... [84]
Utente di Microsoft Office 22/9/20 22:40
... [86]
Silvia 20/9/20 09:49
... [88]

[revised manuscript text omitted]
, if the magma chamber(s) are connected or not to the surface by feeder dykes and conduit. The geometry of the feeding system is prevalent on model outputs with respect to varying rock properties (i.e. Young Modulus) of one order of magnitude. In the case of CVC the use of subsurface homogeneous or stratified lithology not influence much the FEM outputs, being the subsurface geology of the computational domain dominated by carbonates (Unit B).

Beside and beyond the results obtained by analysing the influence of detailed geological and geophysical data, the presented modelling confirms the close to equilibrium state of the volcano, which is the expected stress distribution induced by a feeding system directly connected to the surface.

The complete emptying the upper conduit and part of the shallow magma chamber, as occasionally occurred in the past, originating sub-Plinian and Plinian eruptions, would result in the restoration of the stress arch, which is still a stable stress configuration. Descends that large magnitude, caldera forming eruptions are possible only if the bigger deep magma chamber is also involved and significantly emptied during an eruption.

**Appendices**

**Appendix 1**

E-W gravitational modelling of the CVC domain (stratified lithology) for all configurations investigated. The magnitude and pattern of the principal stress account for a) single magma chamber model (number of nodes: 4426); b) dual magma chamber model (number of nodes: 4161); c) dual magma chamber with conduits model (number of nodes: 3737). The dimension of the deep magma chamber: $2a = 14$ km and $2b = 3.6$ km at 15 km of depth; shallow magma chamber: $2a = 3.5$ km and $2b = 2$ km at 6 km. $\Delta P_e$ and $\Delta P_o$ are equal to 20 MPa for the deep chamber, and 5 MPa for the shallower. Black dotted lines highlight the passage from different stress values. Note that the scales of stress values are different for each panel in order to maximise the simulation details.

**Appendix 2**

E-W gravitational modelling of the CVC domain (stratified lithology) considering an extensional far-field of 5 MPa for all configurations investigated. The magnitude and pattern of the principal stress account for a) single magma chamber model (number of nodes: 4426); b) dual magma chamber model (number of nodes: 4161); c) dual magma chamber with conduits model (number of elements: 3737). The dimension of the deep magma chamber: $2a$ = 14 km and $2b$ = 3.6 km at 15 km of depth; shallow magma chamber: $2a$ = 3.5 km and $2b$ = 2 km at 6 km. $\Delta Pe$ and $\Delta P\rho$ are equal to 20 MPa for the deep chamber, and 5 MPa for the shallower. Black dotted lines highlight the passage from different stress values. The red arrows indicate the direction of the applied far field stress. Note that the scales of stress values are different for each panel in order to maximise the simulation details.

**Code/Data Avaiability**

The LISA code is available at https://lisafea.com/.

**Author's contribution**

SM, RS, AC, GN and GG conceived the study. SM and RS wrote the bulk of the manuscript with the input of all the co-authors. SM and GL compiled the numerical simulations and formulated the adopted methodology. MP and SM carried out the sensitivity analysis. All the authors worked on the interpretation of the results.

**Competing interests:** The authors declare that they have no conflict of interest.

**Acknowledgements:** SM thanks the LISA customer service for the support received. We are deeply grateful to two reviewers and the editor for the critical comments that helped us to improve the clarity and quality of the presentation.

[revised manuscript text omitted]

Silvia 20/9/20 11:13
Silvia 25/9/20 14:36

Silvia 20/9/20 11:14
Silvia 25/9/20 14:36

Silvia 20/9/20 11:15

Silvia 25/9/20 14:36
Silvia 20/9/20 11:15

**Figures**

Figure 1

[Figure]

Figure 2

[Figure]

Figure 3

[Figure]

[Figure]

Figure 5

[Figure]

    Figure 6

[Figure]

[Figure]

Appendix 1

**Appendix 1**

[Figure]

Appendix 2

**Appendix 2**

---

## Author Response (AR3)

**Topical Editor Decision: Publish subject to minor revisions (review by editor)** (05 Oct 2020) by Joachim Gottsmann
Comments to the Author:
Dear Authors,

I have had a look at your revised ms. I am sorry to report that the language issue is far from being resolved. To give an example, the first few sentences of the abstract are full of syntax and grammatical errors and hard to follow.
I hence have to assume that you have not taken my earlier recommendation seriously.

I am giving you one more chance to revise the ms according to my earlier recommendations regarding the use of English. You should consider having the ms proof read by a native speaker or professional editing services.

Should I find the next (third) revision not according to the Journal's publication standards, I will not hesitate to reject the paper altogether.

With best wishes,
Jo Gottsmann
Executive Editor SE

Dear Editor, we are sorry about your last comment. We tried to improve the manuscript as much as possible and we thought was OK, but unfortunately it wasn't. Following your suggestion, the manuscript was sent to a proof read professional editing service (see the attached email). We hope that this revised version is fine. Please note that in the current version the references list is not still formatted because we plan to submit a LaTEX version, if the paper will be accepted for the publication.

We look forward to hearing from you,
Best regards,
Silvia Massaro va ciao
Lucia

---------- Forwarded message ---------
De: **PRS** <accoedj@gmail.com>
Date: vie., 9 de oct. de 2020 a la(s) 05:51
Subject: Proofreading complete: Analysing stress field conditions of the Colima Volcanic Complex (ref. no. 202010-5172859)
To: <lcapra@geociencias.unam.mx>

[Figure]

Proof-Reading-Service.com Ltd, Devonshire
Business Centre, Works Road, Letchworth Garden
City, Hertfordshire, SG6 1GJ, United Kingdom
Office phone: +44(0)20 31 500 431
E-mail: enquiries@proof-reading-service.com
Internet: http://www.proof-reading-service.com
VAT registration number: 911 4788 21
Company registration number: 8391405

Dear Lucia Capra,

We have completed the proofreading you asked for. Please find attached two versions of each document. One is the tracked version, showing all the changes our proofreader has made. You can use the tracking function of Word to accept or reject each change individually. The second is the clean version, which you can use if you do not wish to review the changes we have made. Please note there may be some comments from the proofreader in both versions.

If the clean version still shows corrections please follow the following steps: Press Ctrl+Shift+E and you should see in the task bar a new menu, click on the drop down menu and select 'Final'.

When you are submitting or resubmitting your article to a scientific or academic journal, remember to inform the journal editor in your covering letter that your paper has been professionally proofread. We will be delighted to provide you with verification that your article has been proofread by PRS, so please request a certificate to accompany your paper, especially if the journal editor has already indicated a need for professional proofreading.

If you have further questions about the proofreading, feel free to get in touch.

Thank you for using our service this time. We would be very happy to provide you with further services in the future!

We have recently been experiencing some problems with emails and attachments from some University and business accounts. If you have not heard from us within 2 hours during normal business hours after you have sent your work it is possible that your email has not been received. Please send it from a webmail account such as Hotmail, Yahoo or GMAIL.

Yours sincerely

Emma Taylor

[Figure]

[Figure]

[Figure]

[Figure]

Analysing stress field co...n.docx    Analysing stress field co...n.docx

[revised manuscript text omitted]

---

## Author Response (AR4)

Dear Editor,

we are very happy to accept all your corrections and suggestions about the manuscript. Maybe some errors were due to the proofreading conversion into pdf. We have now prepared the LateX version of the manuscript.

We want also thank the reviewers for having improved the study!
Thanks again for the precious handling of the manuscript.

On the behalf of the authors,

Silvia Massaro